# GNNINTERPRETER: A PROBABILISTIC GENERATIVE MODEL-LEVEL EXPLANATION FOR GRAPH NEURAL NETWORKS

**Xiaoqi Wang**
The Ohio State University
`wang.5502@osu.edu`

**Han-Wei Shen**
The Ohio State University
`shen.94@osu.edu`

## ABSTRACT

Recently, Graph Neural Networks (GNNs) have significantly advanced the performance of machine learning tasks on graphs. However, this technological breakthrough makes people wonder: how does a GNN make such decisions, and can we trust its prediction with high confidence? When it comes to some critical fields, such as biomedicine, where making wrong decisions can have severe consequences, it is crucial to interpret the inner working mechanisms of GNNs before applying them. In this paper, we propose a model-agnostic model-level explanation method for different GNNs that follow the message passing scheme, GNNInterpreter, to explain the high-level decision-making process of the GNN model. More specifically, GNNInterpreter learns a probabilistic generative graph distribution that produces the most discriminative graph pattern the GNN tries to detect when making a certain prediction by optimizing a novel objective function specifically designed for the model-level explanation for GNNs. Compared to existing works, GNNInterpreter is more flexible and computationally efficient in generating explanation graphs with different types of node and edge features, without introducing another blackbox or requiring manually specified domain-specific rules. In addition, the experimental studies conducted on four different datasets demonstrate that the explanation graphs generated by GNNInterpreter match the desired graph pattern if the model is ideal; otherwise, potential model pitfalls can be revealed by the explanation.

## 1 INTRODUCTION

Graphs are widely used to model data in many applications such as chemistry, transportation, etc. Since a graph is a unique non-Euclidean data structure, modeling graph data remained a challenging task until Graph Neural Networks (GNNs) emerged (Hamilton et al., 2017; Cao et al., 2016). As a powerful tool for representation learning on graph data, GNN achieved state-of-the-art performance on various different machine learning tasks on graphs. As the popularity of GNNs rapidly increases, people begin to wonder why one should trust this model and how the model makes decisions. However, the complexity of GNNs prevents humans from interpreting the underlying mechanism in the model. The lack of self-explainability becomes a serious obstacle for applying GNNs to real-world problems, especially when making wrong decisions may incur an unaffordable cost.

Explaining deep learning models on text or image data (Simonyan et al., 2014; Selvaraju et al., 2019) has been well-studied. However, explaining deep learning models on graphs is still less explored. Compared with explaining models on text or image data, explaining deep graph models is a more challenging task for several reasons (Yuan et al., 2020b): (i) the adjacency matrix representing the topological information has only discrete values, which cannot be directly optimized via gradient-based methods (Duval & Malliaros, 2021), (ii) in some application domains, a graph is valid only if it satisfies a set of domain-specific graph rules, so that generating a valid explanation graph to depicts the underlying decision-making process of GNNs is a nontrivial task, and (iii) graph data structure is heterogeneous in nature with different types of node features and edge features, which makes developing a one-size-fits-all explanation method for GNNs to be even more challenging. In this paper, we attempt to interpret the high-level decision-making process of GNNs and to identify potential model pitfalls, by resolving these three challenges respectively.

In recent years, explaining GNNs has aroused great interest, and thus many research works have been conducted. The existing works can be classified into two categories: instance-level explanations (Luo et al., 2020; Ying et al., 2019; Vu & Thai, 2020) and model-level explanations (Yuan et al., 2020a). Instance-level explanation methods focus on explaining the model prediction for a given graph instance, whereas model-level explanation methods aim at understanding the general behavior of the model not specific to any particular graph instance. If the ultimate goal is to examine the model reliability, one will need to examine many instance-level explanations one by one to draw a rigorous conclusion about the model reliability, which is cumbersome and time-consuming. Conversely, the model-level explanation method can directly explain the high-level decision-making rule inside the blackbox (GNN) for a target prediction, which is less time-consuming and more informative regarding the trustworthiness of the GNNs. Besides, it has been shown that any instance-level explanation method would fail to provide a faithful explanation for a GNN that suffers from the bias attribution issue (Faber et al., 2021), while the model-level explanation methods can not only provide a faithful explanation for this case but also diagnose the bias attribution issue.

Even though the model-level explanation methods for GNNs have such advantages, they are much less explored than the instance-level explanation methods. In this paper, we propose a probabilistic generative model-level explanation method for explaining GNNs on the graph classification task, called GNNInterpreter. It learns a generative explanation graph distribution that represents the most discriminative features the GNN tries to detect when making a certain prediction. Precisely, the explanation graph distribution is learned by optimizing a novel objective function specifically designed for the model-level explanation of GNNs such that the generated explanation is faithful and more realistic regarding the domain-specific knowledge. More importantly, GNNInterpreter is a general approach to generating explanation graphs with different types of node features and edge features for explaining different GNNs which follows the message-passing scheme. We quantitatively and qualitatively evaluated the efficiency and effectiveness of GNNInterpreter on four different datasets, including synthetic datasets and public real-world datasets. The experimental results show that GNNInterpreter can precisely find the ideal topological structure for the target prediction if the explained model is ideal, and reveal the potential pitfalls in the model decision-making process if there are any. By identifying these potential model pitfalls, people can be mindful when applying this model to unseen graphs with a specific misleading pattern, which is especially important for some fields in which making wrong decisions may incur an unaffordable cost. Compared with the current state-of-the-art model-level explanation method for GNNs, XGNN (Yuan et al., 2020a), the quantitative and qualitative evaluation result shows that the explanation graphs generated by GNNInterpreter are more representative regarding the target class than the explanations generated by XGNN. Additionally, GNNInterpreter has the following advantages compared with XGNN:

- GNNInterpreter is a more general approach that can generate explanation graphs with different types of node features and edge features, whereas XGNN cannot generate graphs with continuous node features or any type of edge features.

- By taking advantage of the special design of our objective function, GNNInterpreter is more flexible in explaining different GNN models without the need of having domain-specific knowledge for the specific task, whereas XGNN requires domain-specific knowledge to manually design the reward function for the reinforcement learning agent.

- GNNInterpreter is more computationally efficient. The time complexity for GNNInterpreter is much lower than training a deep reinforcement learning agent as in XGNN. Practically, it usually only takes less than a minute to explain one class for a GNN model.

- GNNInterpreter is a numerical optimization approach without introducing another blackbox to explain GNNs, unlike XGNN which trains a deep learning model to explain GNNs.

## 2 RELATED WORK

**Graph Neural Network.** GNNs have achieved remarkable success in different machine learning tasks, including graph classification (Lee et al., 2018), node classification (Xu et al., 2019), and link prediction (Zhang & Chen, 2018). There exists a variety of GNN models (Kipf & Welling, 2017; Gilmer et al., 2017; Veličković et al., 2018), but they often share a common idea of message passing as described in section 3. In this paper, we will focus on interpreting the high-level decision-making process of graph classifiers and diagnosing their potential model pitfall.

**Instance-Level Explanation of GNN.** Encouraged by the success of explaining deep neural networks on image or text data, a multitude of instance-level explanation methods of GNNs have been proposed in the past few years. Their goal is to explain why a particular prediction is being made for a specific input, which is substantially different than our explanation goal. According to a recent survey (Wu et al., 2022), most instance-level explanation methods can be classified into six categories: gradient-based methods(Baldassarre & Azizpour, 2019), perturbation-based methods (Yuan et al., 2021), decomposition methods (Schwarzenberg et al., 2019), surrogate methods (Duval & Malliaros, 2021), generation-based methods(Shan et al., 2021; Lin et al., 2021; 2022), and counterfactual-based methods. Similar to our method, PGExplainer(Luo et al., 2020) also adopts Gumbel-Softmax trick(Maddison et al., 2016) to learn the explanation distribution. However, we learn a model-level explanation distribution obtained by gradient ascent to capture the class-specific discriminative features, while they learn an instance-level explanation distribution predicted by an MLP to identify the important nodes and features toward the GNN prediction for a given graph. In addition to these six categories, some instance-level explanation methods do not fall into any one of the categories. For instance, Veyrin-Forrer et al. (2021) mines the activation pattern in the hidden layers to provide instance-level explanations and summarize the different features of graphs in the training data.

**Model-Level Explanation of GNN.** Model-level explanation methods aim to explain the general behavior of a model without respect to any specific input. The current state-of-the-art model-level explanation method for GNNs is XGNN (Yuan et al., 2020a). Similar to most of the model-level explanation methods for other deep learning models (Simonyan et al., 2014), XGNN and GNNInterpreter share a common goal of generating explanation graphs to maximize the likelihood of the target prediction being made by the GNN model, but with fundamentally different approaches. XGNN trains a deep reinforcement learning model to generate the explanation graphs, while we employ a numerical optimization approach to learn the explanation graph via continuous relaxation. To facilitate the validity of the explanation graph, XGNN incorporates hand-crafted graph rules into the reward function so that the reinforcement learning agent can consider these rules when providing explanations. However, manually specifying these graph rules is cumbersome and requires domain-specific knowledge about the task. XGNN has another limitation that it cannot generate explanation graphs with any type of edge features or continuous node features.

## 3 BACKGROUND

**Notations.** A graph is represented as $G = (\mathbf{V}, \mathbf{E})$, where $\mathbf{V}$ and $\mathbf{E}$ are the sets of nodes and edges. Besides, the number of edges and nodes are denoted as $M$ and $N$, respectively. The topological information of a graph is described by an adjacency matrix $\mathbf{A} \in \{0, 1\}^{N \times N}$ where $a_{ij} = 1$ if there exists an edge between node $i$ and node $j$, and $a_{ij} = 0$ otherwise. In addition, the node feature matrix $\mathbf{X} \in \mathbb{R}^{N \times k_V}$ and edge feature matrix $\mathbf{Z} \in \mathbb{R}^{M \times k_E}$ represent the features for $N$ nodes and $M$ edges.

**Graph Neural Networks.** In general, the high-level idea of GNN is message passing. For each layer, it aggregates information from neighboring nodes to learn the representation of each node. For hidden layer $i$, the message passing operation can be described as $\mathbf{H}^i = f(\mathbf{H}^{i-1}, \mathbf{A}, \mathbf{Z})$ and $\mathbf{H}^0 = \mathbf{X}$, where $\mathbf{H}^i \in \mathbb{R}^{N \times F^i}$ is the hidden node representation output from $i^{th}$ layer and $F^i$ is the output feature dimension. The propagation rule $f$ can be decomposed into three components: computing the messages according to the node embedding at the previous layer $\mathbf{H}^{i-1}$, aggregating the messages from neighboring nodes, and updating the hidden representation $\mathbf{H}^i$ for each node based upon the aggregated message. It is worth mentioning that GNNInterpreter can explain different GNNs that follow this message-passing scheme.

## 4 GNNINTERPRETER

In this paper, we propose a model-level explanation method, GNNInterpreter, which is capable of disclosing the high-level decision-making process of the model for the purpose of examining model reliability. GNNInterpreter is a numerical optimization approach that is model-agnostic for explaining various GNNs following the message-passing scheme. It provides explanation graphs with either discrete or continuous edge features and node features. Besides, we design a novel objective function that considers both the faithfulness and validity of the explanation graphs. Due to the

discreteness nature of graph data, we learn a generative explanation graph distribution by adopting the Gumbel-Softmax trick (Maddison et al., 2016).

## 4.1 Learning Objective

To reveal the high-level decision-making process of the model, one effective approach is to construct an explanation graph that can trigger a specific response from the model as much as possible. Similar to most of the model-level explanation methods for other deep learning models (Simonyan et al., 2014; Nguyen et al., 2015), the explanation for GNNs can be obtained by maximizing the likelihood of the explanation graph $G$ being predicted as a target class by the GNN model. However, solely maximizing the likelihood without constraints may not necessarily result in meaningful explanations. This is because a local maxima in the objective landscape may lie outside of the training data distribution. The explanation yielded from this type of maxima is mathematically legitimate but may provide no meaningful insights on how the model actually reasons over the true data distribution. Especially when there exists domain-specific knowledge that restricts the validity of the data, the explanations are expected to stay inside the hard boundary of the true data distribution. XGNN (Yuan et al., 2020a) addresses this problem by manually specifying a set of graph rules and evaluating the validity of explanation graphs according to the specified rules as a part of the reward function for the reinforcement learning agent. The essential goal is to confine the distribution of the explanation graphs to the domain-specific boundary. Nevertheless, manually specifying rules not only is tedious and time-consuming, but also requires domain expertise which is not always available. Moreover, in many cases, the rules over data distribution can not even be programmatically specified. Therefore, we propose to leverage the abstract knowledge learned by the GNN itself to prevent the explanation from deviating from the true data distribution. This can be achieved by maximizing the similarity between the explanation graph embedding and the average embedding of all graphs from the target class in the training set. Thus, we mathematically formulate our learning objective as follows,

$$\max_G L(G) = \max_{\mathbf{A}, \mathbf{Z}, \mathbf{X}} L(\mathbf{A}, \mathbf{Z}, \mathbf{X}) = \max_{\mathbf{A}, \mathbf{Z}, \mathbf{X}} \phi_c(\mathbf{A}, \mathbf{Z}, \mathbf{X}) + \mu \mathrm{sim}_{\cos}(\psi(\mathbf{A}, \mathbf{Z}, \mathbf{X}), \bar{\psi}_c) \tag{1}$$

where $L$ is the objective function; $\phi_c$ is the scoring function before Softmax corresponding to the class of interest $c$, predicted by the explained GNN; $\psi$ is the graph embedding function of the explained GNN; $\bar{\psi}_c$ is the average graph embedding for all graphs belonging to class $c$ in the training set; $\mathrm{sim}_{\cos}$ denotes the cosine similarity; $\mu$ is a hyper-parameter representing the weight factor.

Given the learning objective defined above, one possible approach to obtain the explanation graph is that we can adopt the gradient ascent to iteratively update the explanation graph $G$ toward maximizing the learning objective. However, this approach cannot be directly applied here because the discrete adjacency matrix $\mathbf{A}$ encoding the graph topology is non-differentiable, namely $\nabla_{\mathbf{A}} L(\mathbf{A}, \mathbf{Z}, \mathbf{X})$ does not exist. Therefore, in order to adopt the gradient ascent method, $\mathbf{A}$ needs to be relaxed to a continuous variable. In addition, if the training graphs of the GNN model have discrete edge features or discrete node features, then $\mathbf{Z}$ and $\mathbf{X}$ are also required to be relaxed to the continuous domain such that $\nabla_{\mathbf{Z}} L(\mathbf{A}, \mathbf{Z}, \mathbf{X})$ and $\nabla_{\mathbf{X}} L(\mathbf{A}, \mathbf{Z}, \mathbf{X})$ exist. Given that the continuous edge features and continuous node features do not require any special accommodation to adopt the gradient ascent, we will only discuss how to learn a continuously relaxed explanation graph with discrete edge features and discrete node features in the following sections.

## 4.2 Learning Generative Graph Distribution with Continuous Relaxation

To learn a probabilistic generative graph distribution, from which the graph drawn maximizes the learning objective, two assumptions have been made (see the detailed justification of the assumptions in Appendix C). The first assumption is that the explanation graph is a Gilbert random graph (Gilbert, 1959), in which every possible edge occurs independently with probability $0 < p < 1$. The second assumption is that the features for each node and edge are independently distributed. Given these two assumptions, the probability distribution of the graph $G$, a random graph variable, can be factorized as

$$P(G) = \prod_{i \in \mathbf{V}} P(x_i) \cdot \prod_{(i,j) \in \mathbf{E}} P(z_{ij}) P(a_{ij}) \tag{2}$$

where $a_{ij} = 1$ when there exists an edge between node $i$ and $j$ and $a_{ij} = 0$ otherwise, $z_{ij}$ denotes the edge feature between node $i$ and $j$, and $x_i$ represents the node feature of node $i$.

Intuitively, $a_{ij}$ can be modeled as a Bernoulli distribution, $a_{ij} \sim \text{Bernoulli}(\theta_{ij})$, in which the probability of success $\theta_{ij}$ represents the probability of an edge existing between node $i$ and $j$ in $G$. For the discrete edge feature $z_{ij}$ and the discrete node feature $x_i$, we can assume that $z_{ij}$ and $x_i$ follow a Categorical distribution, a generalized Bernoulli distribution, with the parameter specifying the probability of each possible outcome for the categorical node feature and the categorical edge feature. Formally, $z_{ij} \sim \text{Categorical}(\mathbf{q}_{ij})$ and $x_i \sim \text{Categorical}(\mathbf{p}_i)$, where $\|\mathbf{q}_{ij}\|_1 = 1$ and $\|\mathbf{p}_{ij}\|_1 = 1$. We define the number of edge categories $k_E = \dim(\mathbf{q}_{ij})$ and the number of node categories $k_V = \dim(\mathbf{p}_i)$. Thus, to learn the probabilistic distribution for explanation graphs, we can rewrite the learning objective as

$$\max_G L(G) = \max_{\boldsymbol{\Theta},\mathbf{Q},\mathbf{P}} \mathbb{E}_{G \sim P(G)}[\phi_c(\mathbf{A},\mathbf{Z},\mathbf{X}) + \mu\text{sim}_{\cos}(\psi(\mathbf{A},\mathbf{Z},\mathbf{X}), \bar{\psi}_c)]. \quad (3)$$

To adopt gradient ascent over the learning objective function with respect to a graph with the discrete edge feature $z_{ij}$ and the discrete node feature $x_i$, we relax $a_{ij}$, $z_{ij}$ and $x_i$ to continuous random variables $\tilde{a}_{ij}$, $\tilde{\mathbf{z}}_{ij}$ and $\tilde{\mathbf{x}}_i$, respectively. Specifically, $a_{ij}$ is relaxed to $\tilde{a}_{ij} \in [0,1]$; $z_{ij}$ is relaxed to $\tilde{\mathbf{z}}_{ij} \in [0,1]^{k_E}, \|\tilde{\mathbf{z}}_{ij}\|_1 = 1$; and $x_i$ is relaxed to $\tilde{\mathbf{x}}_i \in [0,1]^{k_V}, \|\tilde{\mathbf{x}}_i\|_1 = 1$. Given that the Concrete distribution (Maddison et al., 2016) is a family of continuous versions of Categorical random variables with closed-form density, the distribution of continuously relaxed $\tilde{a}_{ij}$, $\tilde{z}_{ij}$ and $\tilde{x}_i$ can be written as

$$\begin{cases} \tilde{a}_{ij} \sim \text{BinaryConcrete}(\omega_{ij}, \tau_a), & \text{for } \tilde{a}_{ij} \in \tilde{\mathbf{A}} \text{ and } \omega_{ij} \in \boldsymbol{\Omega} \\ \tilde{z}_{ij} \sim \text{Concrete}(\boldsymbol{\eta}_{ij}, \tau_z), & \text{for } \tilde{z}_{ij} \in \tilde{\mathbf{Z}} \text{ and } \boldsymbol{\eta}_{ij} \in \mathbf{H} \\ \tilde{x}_i \sim \text{Concrete}(\boldsymbol{\xi}_i, \tau_x), & \text{for } \tilde{x}_i \in \tilde{\mathbf{X}} \text{ and } \boldsymbol{\xi}_i \in \boldsymbol{\Xi} \end{cases} \quad (4)$$

where $\tau \in (0,\infty)$ is the hyper-parameter representing the temperature, $\omega_{ij} = \log(\theta_{ij}/(1-\theta_{ij}))$, $\eta_{ij} = \log \mathbf{q}_{ij}$, and $\xi_i = \log \mathbf{p}_i$. As the temperature approaches zero, the Concrete distribution is mathematically equivalent to the Categorical distribution.

However, in order to learn the generative explanation graph distribution, the sampling function for sampling $\tilde{a}_{ij}$, $\tilde{z}_{ij}$ and $\tilde{x}_i$ from the Concrete distribution is required to be differentiable. Thus, the reparametrization trick is adopted to approximate the sampling procedure of $\tilde{a}_{ij}$, $\tilde{z}_{ij}$ and $\tilde{x}_i$ by a differentiable function. Given an independent random variable $\epsilon \sim \text{Uniform}(0,1)$, the sampling function with the reparameterization trick is as follows,

$$\begin{cases} \tilde{a}_{ij} = \text{sigmoid}\left((\omega_{ij} + \log \epsilon - \log(1-\epsilon))/\tau_a\right) \\ \tilde{z}_{ij} = \text{Softmax}\left((\eta_{ij} - \log(-\log \epsilon))/\tau_z\right) \\ \tilde{x}_i = \text{Softmax}\left((\xi_{ij} - \log(-\log \epsilon))/\tau_x\right) \end{cases} \quad (5)$$

Given this reparameterization trick, we can draw a sample $\tilde{a}_{ij}$ as an approximation of $a_{ij}$, where the probability of $\tilde{a}_{ij}$ to be close to 1 is the same as the probability of $a_{ij}$ to equal 1 and vice versa. Specifically, $P(\tilde{a}_{ij} \to 1) = 1 - P(\tilde{a}_{ij} \to 0) = P(a_{ij} = 1) = \theta_{ij} = \text{sigmoid}(\omega_{ij})$. Similarly, $\tilde{z}_{ij}$ and $\tilde{x}_i$ drawn from the Concrete distribution as in Equation 5 is an approximation of $z_{ij}$ and $x_i$. Therefore, the learning objective with continuous relaxation and reparameterization trick can be approximated by the Monte Carlo method with $K$ samples,

$$\max_{\boldsymbol{\Theta},\mathbf{Q},\mathbf{P}} \mathbb{E}_{G \sim P(G)}[L(\mathbf{A},\mathbf{Z},\mathbf{X})] \approx \max_{\boldsymbol{\Omega},\mathbf{H},\boldsymbol{\Xi}} \mathbb{E}_{\epsilon \sim U(0,1)}[L(\tilde{\mathbf{A}}, \tilde{\mathbf{Z}}, \tilde{\mathbf{X}})] \approx \max_{\boldsymbol{\Omega},\mathbf{H},\boldsymbol{\Xi}} \frac{1}{K}\sum_{k=1}^{K} L(\tilde{\mathbf{A}}, \tilde{\mathbf{Z}}, \tilde{\mathbf{X}}). \quad (6)$$

## 4.3 REGULARIZATION

In order to facilitate optimization and ensure that the explanation graphs are constrained to desired properties, we employ three types of regularization on the latent parameters. Firstly, we apply $L_1$ and $L_2$ regularization on all latent parameters during training to avoid gradient saturation and to help escape from local maxima. Secondly, to encourage the explanation to be as succinct as possible, we impose a budget penalty to limit the size of the explanation graph. Lastly, the connectivity incentive term is introduced to encourage neighboring edge correlation. The equations and detailed explanations for all regularization terms are presented in Appendix A. Putting everything together, the detailed algorithm of GNNInterpreter is shown in algorithm 1.

---

**Algorithm 1:** Generate Explanation Graphs Using GNNInterpreter

---

1   Calculate the average graph embedding $\bar{\psi}_c$ from training data and initialize latent parameters $\mathbf{\Omega}$, $\mathbf{H}$, and $\mathbf{\Xi}$.
2   **while** not converged **do**
3     **for** $k \leftarrow 1 \dots K$ **do**
4        Using the reparameterization trick, sample $\tilde{\mathbf{A}}$, $\tilde{\mathbf{Z}}$, $\tilde{\mathbf{X}}$ with Equation 5.
5        Obtain the class score $s_c^{(k)} \leftarrow \phi_c(\tilde{\mathbf{A}}, \tilde{\mathbf{Z}}, \tilde{\mathbf{X}})$ and the graph embedding $\mathbf{h}_c^{(k)} \leftarrow \psi_c(\tilde{\mathbf{A}}, \tilde{\mathbf{Z}}, \tilde{\mathbf{X}})$.
6     Calculate $L \leftarrow \frac{1}{K} \sum_{k=1}^{K} \left[ s_c^{(k)} + \mu \mathrm{sim}_{\cos}(\mathbf{h}_c^{(k)}, \bar{\psi}_c) \right]$ with the regularization terms.
7     Calculate $\nabla_{\mathbf{\Omega}} L$, $\nabla_{\mathbf{H}} L$, and $\nabla_{\mathbf{\Xi}} L$, then update $\mathbf{\Omega}$, $\mathbf{H}$, and $\mathbf{\Xi}$ using gradient ascent.
8   Let $\mathbf{\Theta} = \mathrm{sigmoid}(\mathbf{\Omega})$, $\mathbf{P} = \mathrm{Softmax}(\mathbf{\Xi})$, $\mathbf{Q} = \mathrm{Softmax}(\mathbf{H})$.
9   **return** $G = (\mathbf{A}, \mathbf{Z}, \mathbf{X})$, where $\mathbf{A} \sim \mathrm{Bernoulli}(\mathbf{\Theta})$, $\mathbf{Z} \sim \mathrm{Categorical}(\mathbf{Q})$, $\mathbf{X} \sim \mathrm{Categorical}(\mathbf{P})$.

---

## 4.4 The Generalizability of GNNInterpreter

GNNInterpreter can generate explanation graphs with different types of node and edge features as needed. If the node or edge feature is continuous, $\nabla_{\mathbf{X}} L(\mathbf{A}, \mathbf{Z}, \mathbf{X})$ and $\nabla_{\mathbf{Z}} L(\mathbf{A}, \mathbf{Z}, \mathbf{X})$ naturally exist, only the adjacency matrix $\mathbf{A}$ need to be relaxed as shown in Equation 4. In Addition, unlike XGNN, which only works with a single categorical node feature, the number of edge and node features are not limited in GNNInterpreter. Generally speaking, GNNInterpreter can explain all common GNNs following the message passing scheme, in which messages are aggregated from their neighbors to update the current node embedding. To adapt message passing for GNNInterpreter, messages are passed between every pair of nodes, but each message is weighted by $\theta_{ij} = \mathrm{sigmoid}(\omega_{ij})$. Thus, if $\theta_{ij}$ approaches 1, node $i$ and node $j$ are highly likely to be neighbors. With this small modification, GNNInterpreter can be easily applied to different message-passing-based GNNs.

## 5 Experimental Study

To comprehensively assess the effectiveness of GNNInterpreter, we carefully design experimental studies conducted on 4 datasets to demonstrate our flexibility of generating explanation graphs with various feature types and manifest our capability of explaining different GNNs. Specifically, we train a GNN classifier for each dataset and adopt GNNInterpreter to explain. In addition, we quantitatively and qualitatively compare the efficacy of GNNInterpreter with XGNN (Yuan et al., 2020a). Details about the experiments and synthetic datasets generation are presented in Appendix G,H,I, and J.

Table 1: The statistics of datasets and some technical details about their corresponding GNN models.

| Dataset | # of Classes | Node Features | Edge Features | Average # of Edges | Average # of Nodes | GNN Type | GNN Test Accuracy |
|---|---|---|---|---|---|---|---|
| **MUTAG (Kersting et al., 2016)** | 2 | ✓ | | 19.79 | 17.93 | GCN (Kipf & Welling, 2017) | 0.9468 |
| **Cyclicity** | 3 | | ✓ | 52.76 | 52.51 | NNConv (Gilmer et al., 2017) | 0.9921 |
| **Motif** | 5 | ✓ | | 77.36 | 57.07 | GCN (Kipf & Welling, 2017) | 0.9964 |
| **Shape** | 5 | | | 71.86 | 30.17 | GCN (Kipf & Welling, 2017) | 0.9725 |

## 5.1 Datasets

**Synthetic Dataset.** We synthetically generate 3 datasets which are Cyclicity, Motif, and Shape (see Table 1). (1) In **Cyclicity**, the graphs have a categorical edge feature with 2 possible values: green or red. There are 3 class labels in Cyclicity, which are Red-Cyclic graphs, Green-Cyclic graphs, and Acyclic graphs. Specifically, a graph will be labeled as a green/red cyclic graph only if it is cyclic and the edges forming that cycle are all green/red. If a graph is categorized as Acyclic, then it either does not contain a cycle at all or contains a cycle but with a mixture of green and red edges. (2) The graphs in **Motif** are labeled according to whether they contain a certain motif. There are 4 motifs (i.e., House, House-X, Complete-4, and Complete-5) will possibly appear in the graphs and each motif corresponds to one class label. There is another class that contains the graphs without any special motif. Besides, each node in the graphs has a categorical node feature with 5 possible values: red, orange, green, blue, and purple. (3) **Shape** contains 5 classes of graphs: Lollipop, Wheel, Grid, Star, and Others. In all the classes except the Others class, graphs are generated with the corresponding shape. For the Others class, graphs are randomly generated without any special topology.

**Real-world Dataset. MUTAG** (Kersting et al., 2016) is a real-world dataset with 2 classes: Mutagen and Nonmutagen. Each molecule graph is categorized according to its mutagenic effect (Debnath

et al., 1991). Given the fact that the molecules with more $NO_2$ are more likely to mutate, the graphs in the Mutagen class tend to have more $NO_2$ than those in the Nonmutagen class. Besides, the Mutagen class and the Nonmutagen class share a common feature which is rings of carbon atoms.

## 5.2 RESULTS

To investigate whether we can truly explain the high-level decision-making process, we evaluated GNNInterpreter both quantitatively and qualitatively. Following XGNN (Yuan et al., 2020a), the class probability of the explanation graphs predicted by the GNN is adopted as the quantitative metric since GNNInterpreter and XGNN share a common objective of maximizing the target class score (Table 2). To qualitatively evaluate GNNInterpreter, we present one explanation graph per class for all datasets in Figure 1, which are all predicted as the target class with the class probability of 1. Due to the page limit, more explanation graphs per class per dataset are presented in Appendix F.

Table 2: The quantitative results for 4 datasets. As the quantitative metric, we compute the average class probability of 1000 explanation graphs and the standard deviation of them for every class in all 4 datasets. In addition, the average training time per class of training 100 different GNNInterpreter and XGNN models, is also included for efficiency evaluation.

| Dataset [Method] | Predicted Class Probability by GNN | | | | Training Time Per Class |
|---|---|---|---|---|---|
| MUTAG [XGNN] | Mutagen
$0.986 \pm 0.057$ | Nonmutagen
$0.991 \pm 0.083$ | | | 128 s |
| MUTAG [Ours] | Mutagen
$1.000 \pm 0.000$ | Nonmutagen
$1.000 \pm 0.000$ | | | 12 s |
| Cyclicity [Ours] | Red Cyclic
$1.000 \pm 0.000$ | Green Cyclic
$1.000 \pm 0.000$ | Acyclic
$1.000 \pm 0.000$ | | 49 s |
| Motif [Ours] | House
$0.918 \pm 0.268$ | House-X
$0.999 \pm 0.032$ | Complete-4
$1.000 \pm 0.000$ | Complete-5
$0.998 \pm 0.045$ | 83 s |
| Shape [Ours] | Lollipop
$0.742 \pm 0.360$ | Wheel
$0.989 \pm 0.100$ | Grid
$0.996 \pm 0.032$ | Star
$1.000 \pm 0.000$ | 24 s |

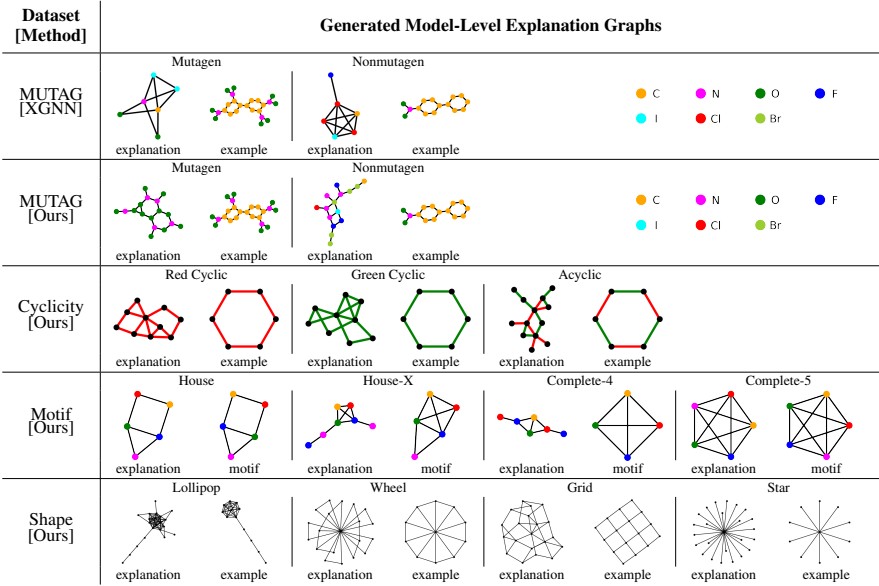

Figure 1: The qualitative results for 4 datasets. For each class in all datasets, the explanation graph with the class probability of 1 predicted by the GNNs is displayed on the left; as a reference, the example graph selected from the training data of the GNNs or the motif is displayed on the right. The different colors in the nodes and edges represent different values in the node feature and edge feature.

Since XGNN is the current state-of-the-art model-level explanation method for GNNs, we compare GNNInterpreter with XGNN quantitatively and qualitatively. However, given that XGNN cannot generate explanation graphs with edge features, XGNN is not applicable to the Cyclicity dataset. For Motif and Shape, we attempted to adopt XGNN to explain the corresponding GNNs, but the

explanation results were not acceptable even after our extensive efforts of trial-and-error on XGNN. Therefore, for a fair comparison, we only present the evaluation results of XGNN on the GNN trained on MUTAG, with the same hyper-parameter settings for MUTAG as they discussed in their paper. In addition to these 4 datasets, we also conducted experiments on the Is_Acyclic dataset synthesized by XGNN authors, but we failed to reproduce similar results claimed in the XGNN paper. The details of our comparative study on the Is_Acyclic dataset against XGNN can be found in subsection D.2.

**MUTAG.** The quantitative evaluation in Table 2 shows that all explanation graphs generated by GNNInterpreter are classified correctly with 100% probability for both classes with zero variance, which indicates that we can consistently generate the most representative explanation graphs in terms of the explained GNN for all three classes. On the contrary, the GNN is more hesitant or diffident when classifying the XGNN explanation graphs for both classes. Speaking of the time efficiency, for explaining the same GNN model trained on MUTAG, *we can generate an explanation graph about 10 times faster than XGNN on average according to Table 2* (see the time complexity analysis of XGNN and GNNInterpreter in Appendix B). In Figure 1, the GNNInterpreter explanation for Mutagen class contains 5 repeated $NO_2$ whereas the GNNInterpreter explanation for Nonmutagen class does not contain $O$ atom at all, which is perfectly aligned with our background knowledge about the MUTAG dataset mentioned in subsection 5.1. In contrast, the XGNN explanation for the Mutagen class only contains a single $NO_2$, which is a less accurate or ideal explanation than ours. In addition, according to our explanation graphs for both classes, they indicate that the GNN trained with MUTAG is likely to suffer from the bias attribution issue (Faber et al., 2021) because the explanation graph for Nonmutagen class does not contain any special pattern but achieve the class probability of 1. This finding is further verified by the additional experiments we conducted (see subsection E.2).

**Cyclicity.** In addition to the good quantitative performance shown in Table 2, the explanation graphs for Red Cyclic and Green Cyclic only contain multiple red cycles and multiple green cycles, respectively; the explanation graph for Acyclic does not contain any cycle with homogeneous edge colors (see Figure 1). These explanations reveal a pattern that the *more occurrences of a discriminative feature for the corresponding class (i.e., red cycle, green cycle, and cycle with heterogeneous edge color, respectively) the more confident the GNN is during the decision-making process*.

**Motif.** The result in Table 2 shows that the explanations for all classes successfully identify the discriminative features perceived by the GNN. In Figure 1, the explanations for House and Complete-5 classes match the exact motif. Even though the explanation graphs for House-X and Complete-4 do not match the exact motif, it still obtains the class probabilities of 1. By analyzing the commonality between the explanations and their corresponding ground truth motifs for House-X and Complete-4 respectively, we speculate the discriminative features the GNN tries to detect for these two classes are the following: in House-X graphs, the orange node and the red node should have 3 neighbors except the purple node, the green node and the blue node should have 4 neighbors in a different color; in Complete-4 graphs, every node should have 3 neighbors in a different color. *Clearly, the GNN with a test accuracy of 0.9964 ( Table 1) perceives the wrong motifs as the House-X and Complete-4, which implies the potential risk of misclassification for unseen graphs with misleading patterns.* The inconsistency between the explanations and the truth motifs raises a red flag that the GNN might be cheating by using some partly correct discriminative features as evidence to make decisions.

**Shape.** In Table 2, the explanations for all the classes except Lollipop achieve a good quantitative performance. For Lollipop, the average class probability of 0.742 is still a reasonably high probability than 0.2 (the baseline probability of random guessing for a classification task with 5 classes). Besides, the explanation graphs for all 4 classes presented in Figure 1 are almost consistent with the desired graph patterns. Nonetheless, it also discloses some interesting findings regarding the high-level decision-making process of the GNN. For Wheel and Grid class, the explanation graphs are not very perfect but with the predicted class probability of 1. Accordingly, we infer the true belief inside the GNN is that: the discriminative features of Wheel graph and Grid graph are a single common vertex shared by multiple triangles and multiple polygons sharing some common edges, respectively. *These might become the potential model pitfalls because, for example, the GNN will be easily tricked by a non-wheel graph in which there exists a common vertex shared by multiple triangles.*

**Verification Study.** Since the ground truth model-level explanation graphs are unknown, the correctness of our analysis regarding these potential GNN pitfalls cannot be directly justified. Therefore, to verify the correctness, we conduct controlled experimental studies to test how the GNN behaves on some carefully chosen input graphs with specific misleading patterns. As mentioned above, we

speculate the discriminative feature the GNN tries to detect for the House-X class is that "orange node and red node should have 3 neighbors except purple node, green node and blue node should have 4 neighbors in different color". We found 9 plausible motifs (including the ground truth House-X) which satisfy this rule (see Figure 2). Given a fixed set of 5000 base graphs, each of the 9 different motifs is attached to the same node of each base graph respectively. Thus, we obtain 9 sets of graphs, each of which contains 5000 graphs corresponding to one of the motifs. For each motif set, the classification results are presented in Table 3. We can see that *motif 1-8 successfully fool the explained GNN because a large portion of the non-House-X graphs are incorrectly classified as House-X.* This verification result strongly supports our qualitative analysis regarding the potential model pitfall for House-X class. The verification studies for more classes are presented in Appendix E.

**Plausible House-X Motifs**

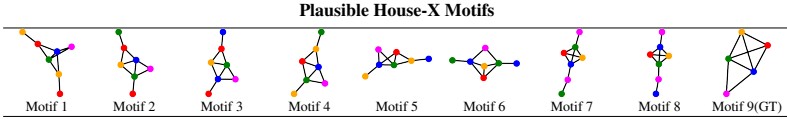

Figure 2: The ground truth class label for motif 1-8 and motif 9 is Others and House-X, respectively.

Table 3: The classification results of plausible House-X motifs.

|  | Motif 1 | Motif 2 | Motif 3 | Motif 4 | Motif 5 | Motif 6 | Motif 7 | Motif 8 | Motif 9 (GT) |
|---|---|---|---|---|---|---|---|---|---|
| **Others** | 1434 | 2251 | 1808 | 1970 | 2183 | 597 | 1682 | 1696 | 10 |
| **House** | 0 | 0 | 0 | 0 | 0 | 0 | 0 | 0 | 0 |
| **House-X** | 3566 | 2749 | 3192 | 3030 | 2817 | 4403 | 3318 | 3304 | 4990 |
| **Complete-4** | 0 | 0 | 0 | 0 | 0 | 0 | 0 | 0 | 0 |
| **Complete-5** | 0 | 0 | 0 | 0 | 0 | 0 | 0 | 0 | 0 |

**Ablation Study.** In many cases, especially when true graphs are constrained with certain domain-specific rules, the second term of the objective function in Equation 1 plays a significant role in generating meaningful explanations. To assess the effectiveness of the second term, we take the Mutagen class from MUTAG dataset as an example to conduct an ablation study on this term. As shown in Figure 3, the explanation without the second term completely deviates from the true data distribution because the ground truth discriminative feature of the mutagen class (i.e., NO2) is completely missing in the explanation without the second term. Even though the explanation without the second term achieves a significantly higher logit, it is difficult to gain meaningful insights from the explanation without the second term. That is to say, the out-of-distribution explanation graph generated without the second term might not provide meaningful insight into how the model actually reasons over the true data distribution, which is what we actually care about.

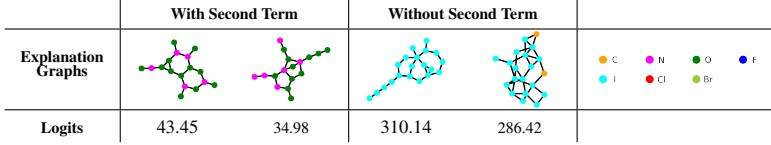

Figure 3: The ablation study of the second term on the mutagen class from MUTAG dataset. For each category (column), we present 2 explanation graphs with the class score before Softmax (logits).

## 6 CONCLUSION

We propose GNNInterpreter, a probabilistic generative model-level explanation method for GNNs, which learns a graph distribution to depict the discriminative and representative pattern perceived by the GNNs for a target class. By optimizing the objective function we designed, GNNInterpreter can generate faithful and realistic explanations, without requiring manually specified domain-specific graph rules. Compared with XGNN, GNNInterpreter is more general and computationally efficient because it works with different types of edge and node features, while having lower time complexity. This makes GNNInterpreter more practical and accessible in real-world applications. More importantly, the experimental results show that GNNInterpreter can not only produce more faithful results than XGNN on the real-life dataset, but also reveal the potential model pitfalls such as the bias attribution issue and misclassification of unseen graphs with misleading graph patterns.

## 7 ACKNOWLEDGEMENT

The work was supported in part by the US Department of Energy SciDAC program DE-SC0021360, National Science Foundation Division of Information and Intelligent Systems IIS-1955764, and National Science Foundation Office of Advanced Cyberinfrastructure OAC-2112606.

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

# Appendix

## Table of Contents

## A    REGULARIZATION

In order to facilitate optimization and ensure that the explanation graphs are constrained to desired properties, we apply multiple regularization terms to the latent parameters.

$L_1$ **and** $L_2$ **Regularization.** $L_1$ and $L_2$ regularizations are widely used to encourage model sparsity and reduce the parameter size. In our case, both regularizations are applied on $\mathbf{\Omega}$ during the optimization to reduce its size, with the purpose of mitigating the saturating gradient problem caused by the sigmoid function in Equation 5. For $k \in \{1, 2\}$, $L_k$ penalty term is defined as

$$R_{L_k} = \|\mathbf{\Omega}\|_k. \tag{7}$$

**Budget Penalty.** Budget penalty is employed in instance-level GNN explanation methods (i.e., PGExplainer (Luo et al., 2020) and GNNExplainer (Ying et al., 2019)) to generate compact and meaningful explanations. For the model-level explanation, the budget penalty can prevent the size of explanation graphs from growing unboundedly with repeated discriminative patterns. The budget penalty is defined as

$$R_b = \mathrm{softplus}\left(\|\mathrm{sigmoid}(\mathbf{\Omega})\|_1 - B\right)^2, \tag{8}$$

where $B$ is the expected maximum number of edges in the explanation graph. *Unlike PGExplainer, we utilize* softplus *instead of* ReLU *to slack the rate of gradient change near the boundary condition. In addition, the penalty is squared to further discourage extremely large graphs.*

**Connectivity Incentive.** Inspired by PGExplainer, we apply a connectivity incentive to encourage explanation graphs to be connected, by minimizing the Kullback–Leibler divergence between the probabilities of each pair of edges that share a common node in the Gilbert random graph,

$$R_c = \sum_{i \in V} \sum_{j,k \in \mathcal{N}(i)} D_{\text{KL}}(P_{ij} \| P_{ik}) \tag{9}$$

where $P_{ij}$ denotes the Bernoulli distribution parameterized by $\text{sigmoid}(\omega_{ij})$. *As opposed to PGExplainer, KL-divergence is chosen over cross-entropy in order to decouple the entropy of individual edge distributions from the connectivity incentive.*

## B  THE TIME COMPLEXITY ANALYSIS OF GNNINTERPRETER AND XGNN

---

**Algorithm 2:** Time Complexity Analysis for GNNInterpreter

---

1  Calculate the average graph embedding $\bar{\psi}_c$ from training data and initialize latent parameters $\mathbf{\Omega}$, $\mathbf{H}$, and $\mathbf{\Xi}$.
2  **while** not converged **do**
3      **for** $k \leftarrow 1...K$ **do**
4          Using the reparameterization trick, sample $\tilde{\mathbf{A}}, \tilde{\mathbf{Z}}, \tilde{\mathbf{X}}$ with Equation 5. $O(2N^2 + N) = O(N^2)$
5          Obtain class score $s_c^{(k)} \leftarrow \phi_c(\tilde{\mathbf{A}}, \tilde{\mathbf{Z}}, \tilde{\mathbf{X}})$ and graph embedding $\mathbf{h}_c^{(k)} \leftarrow \psi_c(\tilde{\mathbf{A}}, \tilde{\mathbf{Z}}, \tilde{\mathbf{X}})$. $O(N^2)$
6      Calculate $L \leftarrow \frac{1}{K} \sum_{k=1}^{K} \left[ s_c^{(k)} + \mu \text{sim}_{\cos}(\mathbf{h}_c^{(k)}, \bar{\psi}_c) \right]$ with the regularization terms. $O(K)$
7      Calculate $\nabla_{\mathbf{\Omega}} L, \nabla_{\mathbf{H}} L$, and $\nabla_{\mathbf{\Xi}} L$, then update $\mathbf{\Omega}$, $\mathbf{H}$, and $\mathbf{\Xi}$ using gradient ascent. $O(N^2)$
8  Let $\mathbf{\Theta} = \text{sigmoid}(\mathbf{\Omega})$, $\mathbf{P} = \text{Softmax}(\mathbf{\Xi})$, $\mathbf{Q} = \text{Softmax}(\mathbf{H})$.
9  **return** $G = (\mathbf{A}, \mathbf{Z}, \mathbf{X})$, where $\mathbf{A} \sim \text{Bernoulli}(\mathbf{\Theta})$, $\mathbf{Z} \sim \text{Categorical}(\mathbf{Q})$, $\mathbf{X} \sim \text{Categorical}(\mathbf{P})$.

---

**GNNInterpreter.** Let $N$ be the number of vertices for the Gilbert random graph distribution, which indicates the maximum number of nodes in the explanation graph. In algorithm 2, for each Monte Carlo sampling process in line 4, it takes $O(N^2)$ to sample $\tilde{\mathbf{A}}$ and $\tilde{\mathbf{Z}}$, and takes $O(N)$ to sample $\tilde{\mathbf{X}}$. To obtain the class score and the graph embedding in line 5, it requires one forward pass through the GNN to be explained, which takes $O(N^2)$ time. Therefore, the inner loop takes $O(3N^2 + N) = O(N^2)$ as a whole. Performing $K$ Monte Carlo sampling will thus take $O(KN^2)$. Then, in line 6, computing loss function takes $O(K)$ for $K$ samples from the inner loop. In the end, performing back-propagation and gradient ascent also takes $O(N^2)$ in line 7. For each iteration, we have $O((K+1)N^2 + K) = O(KN^2)$. Assuming that GNNInterpreter takes $T$ iterations to converge, the time complexity of the entire algorithm is $O(TKN^2)$.

**XGNN.** In comparison, XGNN is a reinforcement learning-based approach for generating explanation graphs. It formulates the graph generator as a policy function that generates one edge at a time and uses the GNN class score as a reward function. It leverages the policy gradient method to optimize the policy function during training. The whole XGNN training procedure runs repeatedly for $T$ episodes. For each episode, $M$ generation steps are performed to generate a graph with at most $M$ edges. A Rollout procedure is performed $R$ times in each generation step. During Rollout, each generation step takes $O(M)$ time for the GNN forward pass to evaluate the reward value, with a maximum of $M$ generation steps. In total, $R$ times of Rollout procedure will take $O(RM^2)$ time. For $M$ generation steps in each episode, the time complexity is $O(RM^3)$. Assuming the total number of episodes is $T$, the time complexity of training an XGNN model is then $O(TRM^3)$.

For a connected explanation graph with $N$ nodes and $M$ edges, $M > N - 1$ always holds. Since $K$ and $R$ are hyper-parameters, we can treat them as constants. Thus, $O(TRM^3) = O(M^3)$ is much higher than $O(TKN^2) = O(N^2)$. As a result, GNNInterpreter outperforms XGNN in terms of time complexity.

## C   Justification of Assumptions Regarding Explanation Graph Distribution

In this section, we would like to justify the two assumptions we made regarding the explanation graph distribution. Our first assumption is that we assume the explanation graph is a Gilbert graph(Gilbert, 1959). Compared with other widely used probability models over graphs (i.e., Erdo-Renyi graph (ERDdS & R&WI, 1960), Rado graph(Rado, 1964), and random dot-product model), Gilbert random graph is more suitable for our cases. First, the Erdo-Renyi graph models the graph in a probability space of a fixed number of edges and nodes. However, for generating the model-level explanation graphs, we do not want to enforce a hard constraint on the number of edges a priori. Instead, we want our explanation method to flexibly learn how many edges should be in the explanation graph. Speaking of the Rado graph, it is a random graph with an infinite number of nodes and edges distributed independently, which will result in an explanation graph with an infinite number of nodes. Apparently, we are unable to visually analyze an explanation graph with an infinite number of nodes. Lastly, the random dot-product graph model is just a generalized version of the Gilbert random graph. Therefore, we decide to model the explanation graph distribution as a Gilbert random graph because it is the most reasonable choice for our purpose.

Our second assumption is assuming features for each node and edge are independently distributed. We chose to make this assumption because this will simplify most of the mathematics as a standard practice in statistics. We acknowledge that this assumption might not always hold, but our explanation method actually takes into account the dependency in a different way while learning the explanation graph distribution. The reason is that the training process will compute the gradient through the message-passing operation inside GNN, and the message-passing operation exchanges information across the whole graph in the forward pass. Therefore, the gradient computed backward will capture the correlation among latent parameters of the independent distributions. Through this mechanism, the dependency among edge distributions, edge feature distributions, and node feature distributions are all taken into account in the learning process. That is to say, the independence assumption we made is only applied to sampling an explanation graph from the learned explanation graph distribution, while the latent parameters of those distributions are actually correlated under the hood.

## D   Additional Evaluation Results

### D.1   Random Baseline

To better demonstrate the effectiveness of our method, we also quantitatively evaluate how the classifier would behave on a set of random graphs, which can serve as the evaluation of a random baseline. For each dataset, we generate 1000 Gabriel random graphs with the same number of nodes as the average number of nodes in Table 4. Then, the average class probability with the standard deviation (predicted by the corresponding classifier) for 1000 random graphs is computed and reported in Table 5. As you can see, for all the classifiers we explained, our explanation graph can obtain a significantly higher class probability than the random graphs on average. It is very interesting to see that, the random graphs are very likely to be classified as the negative class, for the dataset including MUTAG, Cyclicity, Is_Acyclic, and ColorConsistency.

### D.2   Is_Acyclic: Synthetic Dataset from XGNN Paper

In order to conduct a more comprehensive comparison with the current state-of-the-art method (XGNN), we managed to evaluate GNNInterpreter on Is_Acyclic data which is synthetically generated by the XGNN authors and posted at XGNN GitHub[1]. We trained a GCN classifier on this dataset with an accuracy of 1 (see the last row of Table 4) and adopted GNNInterpreter to explain the two classes in Is_Acyclic. The quantitative and qualitative evaluation of GNNInterpreter on Is_Acyclic is conducted in a similar manner of other datasets, which is shown in Table 5 and Figure 4, respectively.

However, it is unfortunate that XGNN could not provide any meaningful explanation for the GCN model we trained on Is_Acyclic (like what we experienced for the Shape and Motif dataset). We spent an extensive amount of time tuning the hyper-parameter of XGNN, but we still could not obtain

---

[1]https://github.com/divelab/DIG/tree/main/dig/xgraph/XGNN

meaningful and acceptable explanation graphs. We also attempted to conduct a comparative study on the GNN model trained by the XGNN authors because its model checkpoint is available in the google drive folder they shared. But unfortunately, the model definition of the GNN model on Is_Acyclic is not available on their GitHub (only the model definition of GNN for MUTAG is provided on their GitHub), so we cannot successfully load the model checkpoint they trained for Is_Acyclic. Even so, the quantitative and qualitative evaluation of XGNN on the GCN model we trained on Is_Acyclic is still presented in Table 5 and Figure 4.

Speaking of the evaluation results, the quantitative result in Table 5 shows that, for both classes, we can consistently generate faithful explanations containing the discriminative pattern of the target class perceived by the GNN. However, the XGNN explanations for both classes are less ideal because the average class probability over 1000 explanation graphs for the Cyclic class is almost zero. This indicates that the explanation graphs generated for both classes will be classified as Acyclic by the GNN, no matter which is the target class to explain. In terms of computational time, we can generate an explanation graph about two times faster than XGNN on average. Qualitatively speaking, our explanation graphs for both Cyclic and Acyclic classes are consistent with the class definition with the expected properties (see Figure 4). Namely, the explanation graph for Cyclic contains cycles, while the explanation graph for Acyclic does not contain any cycles. In contrast, for both classes, XGNN fails to generate an explanation graph beyond two nodes, even after our extensive trail-and-error effort.

Table 4: The statistics of datasets including Is_Acyclic and ColorConsistency, and some technical details about their corresponding GNN models.

| Dataset | # of Classes | Node Features | Edge Features | Average # of Edges | Average # of Nodes | GNN Type | GNN Test Accuracy |
|---|---|---|---|---|---|---|---|
| MUTAG (Kersting et al., 2016) | 2 | ✓ | | 19.79 | 17.93 | GCN (Kipf & Welling, 2017) | 0.9468 |
| Cyclicity | 3 | | ✓ | 52.76 | 52.51 | NNConv (Gilmer et al., 2017) | 0.9921 |
| Motif | 5 | ✓ | | 77.36 | 57.07 | GCN (Kipf & Welling, 2017) | 0.9964 |
| Shape | 5 | | | 71.86 | 30.17 | GCN (Kipf & Welling, 2017) | 0.9725 |
| Is_Acyclic | 2 | | ✓ | 30.04 | 28.46 | GCN (Kipf & Welling, 2017) | 1.0000 |
| ColorConsistency | 2 | ✓ | ✓ | 69.06 | 52.33 | GAT (Veličković et al., 2018) | 0.9963 |

Table 5: The quantitative evaluation results for all 6 datasets including Is_Acyclic. As the quantitative metric, we compute the average class probability of 1000 explanation graphs and the standard deviation of them for the two classes. In addition, the average training time per class of training 100 different GNNInterpreter and XGNN models is also included for efficiency evaluation.

| Dataset [Method] | Predicted Class Probability by GNN | | | | Training Time Per Class |
|---|---|---|---|---|---|
| Is_Acyclic [XGNN] | Cyclic $0.076 \pm 0.000$ | Acyclic $0.927 \pm 0.000$ | | | 45 s |
| Is_Acyclic [Ours] | Cyclic $0.999 \pm 0.001$ | Acyclic $1.000 \pm 0.000$ | | | 20 s |
| Is_Acyclic [Random] | Cyclic $0.143 \pm 0.155$ | Acyclic $0.857 \pm 0.155$ | | | - |
| MUTAG [XGNN] | Mutagen $0.986 \pm 0.057$ | Nonmutagen $0.991 \pm 0.083$ | | | 128 s |
| MUTAG [Ours] | Mutagen $1.000 \pm 0.000$ | Nonmutagen $1.000 \pm 0.000$ | | | 12 s |
| MUTAG [Random] | Mutagen $0.068 \pm 0.251$ | Nonmutagen $0.932 \pm 0.251$ | | | - |
| ColorConsistency [Ours] | Consistent $0.968 \pm 0.110$ | Inconsistent $1.000 \pm 0.000$ | | | 58 s |
| ColorConsistency [Random] | Consistent $0.017 \pm 0.118$ | Inconsistent $0.983 \pm 0.118$ | | | - |
| Cyclicity [Ours] | Red Cyclic $1.000 \pm 0.000$ | Green Cyclic $1.000 \pm 0.000$ | Acyclic $1.000 \pm 0.000$ | | 49 s |
| Cyclicity [Random] | Red Cyclic $0.023 \pm 0.143$ | Green Cyclic $0.015 \pm 0.118$ | Acyclic $0.962 \pm 0.183$ | | - |
| Motif [Ours] | House $0.918 \pm 0.268$ | House-X $0.999 \pm 0.032$ | Complete-4 $1.000 \pm 0.000$ | Complete-5 $0.998 \pm 0.045$ | 83 s |
| Motif [Random] | House $0.000 \pm 0.000$ | House-X $0.000 \pm 0.000$ | Complete-4 $0.000 \pm 0.000$ | Complete-5 $0.000 \pm 0.000$ | - |
| Shape [Ours] | Lollipop $0.742 \pm 0.360$ | Wheel $0.989 \pm 0.100$ | Grid $0.996 \pm 0.032$ | Star $1.000 \pm 0.000$ | 24 s |
| Shape [Random] | Lollipop $0.214 \pm 0.301$ | Wheel $0.000 \pm 0.000$ | Grid $0.151 \pm 0.307$ | Star $0.000 \pm 0.000$ | - |

| Dataset [Method] | Generated Model-Level Explanation Graphs |
|---|---|
| Is_Acyclic[XGNN] | 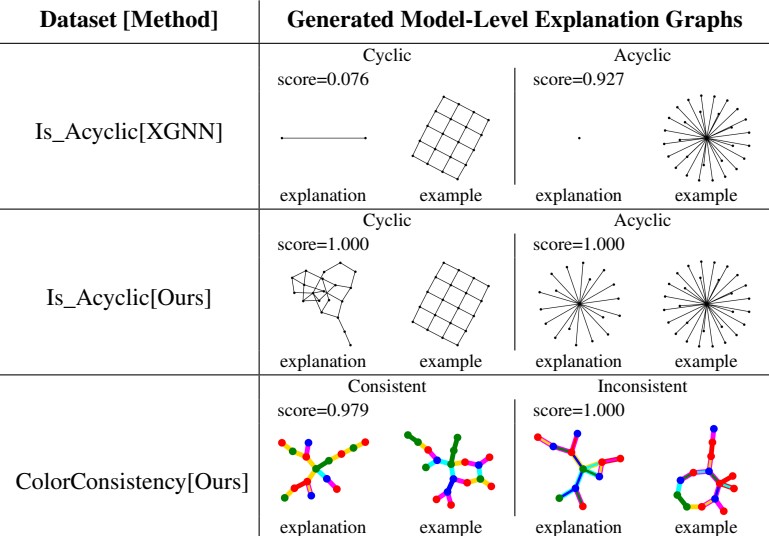 |
| Is_Acyclic[Ours] | |
| ColorConsistency[Ours] | |

Figure 4: The qualitative results for Is_Acyclic and ColorConsistency. As a reference, the example graph selected from the training data of the GNNs is displayed on the right. Also, the class score after Softmax of the target class is presented above each corresponding explanation graph. For ColorConsistency, the edges with inconsistent color is drawn with a colored outline.

## D.3    COLORCONSISTENCY: SYNTHETIC DATA WITH NODE FEATURES AND EDGE FEATURES

With the purpose of further demonstrating the generalizability of GNNInterpreter, we conduct an experimental study on explaining a GAT(Veličković et al., 2018) model trained to classify a synthetic dataset (ColorConsistency) with both node features and edge features. To be specific, ColorConsistency contains two classes: the Consistent class is the graphs with all the node features and edge features has consistent color; and the Inconsistent class is otherwise. The consistency of color is defined as the natural color addition in physics. For example, if node $v$ is in red and node $u$ is in green, then the edge between $u$ and $v$ should be in yellow. If there is one edge in the graph that does not follow this color consistency, then this graph should be categorized into the Inconsistent class. The statistics of ColorConsistency and the test accuracy of the corresponding GAT Classifier are presented in Table 4. Please refer to algorithm 6 for a detailed generation procedure of ColorConsistency.

To assess our effectiveness on explaining the GAT classifier trained on ColorConsistency, we quantitatively and qualitatively evaluate the faithfulness of the generated explanation graph. According to the quantitative result in Table 5, we can consistently generate an explanation graph with discriminative features of the corresponding class because the predicted class probability of our explanation graphs for both classes is close to or equal to 1 on average. Speaking of the qualitative evaluation, the explanation graph per class along with the predicted class probability is presented in Figure 4. The edges with inconsistent color are drawn with a colored outline. As you can see, the explanation graph of the Inconsistent class matches the definition of ground truth Inconsistent graph because every edge all have an inconsistent color. It may indicate that the GNN has correctly learned the discriminative features of the Inconsistent class. The more discriminative features are detected, the more confident GNN is to classify it as the Inconsistent class. However, the explanation graph of the Consistent class does not follow the true definition of "Consistent" so the ground truth label of the Consistent explanation graph should be "Inconsistent". This is a very interesting observation because the GNN test accuracy is as high as 0.9963 (see Table 4). The explanation of the Consistent class reveals a potential risk of misclassification for graphs with few inconsistently colored edges.

# E  A VERIFICATION STUDY OF OUR ANALYSIS ABOUT QUALITATIVE RESULTS

From Figure 1, we can see that the explanation graphs for some classes are different from the true graphs in the training data, which might indicate the potential pitfall of the explained GNN. However, since the ground truth model-level explanation graph does not exist, it is difficult to verify the correctness of our analysis regarding these potential pitfalls. In this section, we conducted the controlled experimental studies for MUTAG, House-X in Motif and Complete-4 in Motif, to verify our high-level interpretation of the explained GNN mentioned in the subsection 5.2.

## E.1  MOTIF

**Complete-4** In subsection 5.2, we infer that the discriminative feature the GNN tries to detect is that "every node should have 3 neighbors in different color". This rule is concluded from our observation of the commonality between our explanation graph and the ground truth motif. We found 8 different motifs (including the ground truth Complete-4) which satisfy this rule, as shown in Figure 5. Given a fixed set of 5000 Rome graphs, these 8 different motifs are attached to the same node of those 5000 Rome graphs. As a result, we obtain 8 sets of graphs, and each set includes 5000 graphs corresponding to a single motif. For each motif set, the average predicted class probability of each class and the total number of graphs that are classified as each class are presented in the Table 6 and Table 7, respectively. We can clearly see that motif 1-6 successfully fool the explained GNN because their predicted class probability of Complete-4 is significantly higher than their predicted class probability of Others. These results indicate that our previous analysis in subsection 5.2 regarding how the GNN makes predictions for Complete-4 is correct. Namely, the GNN trained with the Motif dataset is highly likely to incorrectly classify the graphs containing a misleading pattern identified by the GNNInterpreter, even though the GNN accuracy is as high as 0.9964 (see Table 1). *Therefore, GNNInterpreter can be served as a sanity check for GNNs, especially the ones with high accuracy, and remind people to be cautious when applying GNNs to classify some unseen graphs containing the misleading patterns identified by the GNNInterpreter.*

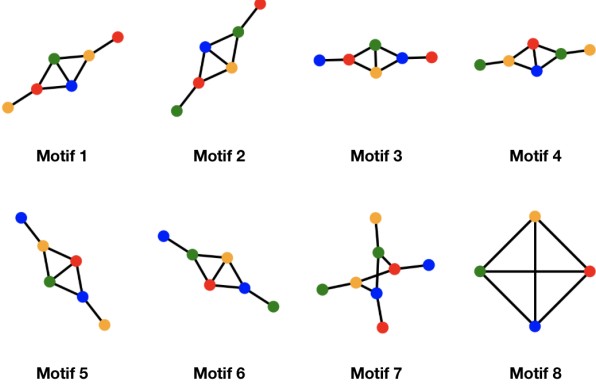

Figure 5: Plausible Complete-4 motifs. The motifs satisfying the rule that every node should have 3 neighbors in a different color. The motif 8 is the ground truth Complete-4 motif. The ground truth class label of motif 1-7 should be the Others class.

Table 6: The quantitative results of plausible Complete-4 motifs. For each motif set (column), the average predicted class probability of each class (row) is computed by averaging over 5000 graphs containing the corresponding motif.

|  | Motif 1 | Motif 2 | Motif 3 | Motif 4 | Motif 5 | Motif 6 | Motif 7 | Motif 8 (GT) |
|---|---|---|---|---|---|---|---|---|
| **Others** | 0.4123 | 0.4064 | 0.4122 | 0.4033 | 0.4116 | 0.4019 | 0.9974 | 0.0066 |
| **House** | 0.0000 | 0.0000 | 0.0000 | 0.0000 | 0.0000 | 0.0000 | 0.0021 | 0.0000 |
| **House-X** | 0.0002 | 0.0006 | 0.0004 | 0.0007 | 0.0003 | 0.0002 | 0.0001 | 0.0000 |
| **Complete-4** | 0.5875 | 0.593 | 0.5874 | 0.596 | 0.5881 | 0.5978 | 0.0004 | 0.9934 |
| **Complete-5** | 0.0000 | 0.0000 | 0.0000 | 0.0000 | 0.0000 | 0.0000 | 0.0001 | 0.0000 |

Table 7: The classification results of plausible Complete-4 motifs. For each motif set (column), the total number of graphs that are classified as classes (row) is presented in this table. The sum of all columns should be 5000 because that is the total number of graphs in each motif set.

| | Motif 1 | Motif 2 | Motif 3 | Motif 4 | Motif 5 | Motif 6 | Motif 7 | Motif 8 (GT) |
|---|---|---|---|---|---|---|---|---|
| **Others** | 2030 | 1983 | 2016 | 1985 | 2031 | 1966 | 4994 | 0 |
| **House** | 0 | 0 | 0 | 0 | 0 | 0 | 4 | 0 |
| **House-X** | 1 | 3 | 1 | 4 | 1 | 1 | 0 | 0 |
| **Complete-4** | 2969 | 3014 | 2983 | 3011 | 2986 | 3033 | 2 | 5000 |
| **Complete-5** | 0 | 0 | 0 | 0 | 0 | 0 | 0 | 0 |

**House-X** In subsection 5.2, we speculate the discriminative feature perceived by the explained GNN is that "orange node and red node should have 3 neighbors except purple node, green node and blue node should have 4 neighbors in different color". This discriminative feature is also deduced from the commonality between our explanation for House-X and the ground truth motif. The 9 motifs satisfying this rule are shown in Figure 6. The controlled experimental study for House-X class is conducted in a similar manner as for Complete-4 class. The quantitative results for each motif set in Table 8 and Table 9 show that all the plausible motifs satisfying the rule are highly likely to be classified as House-X, no matter what are their corresponding ground truth labels. Specifically, a large number of graphs from motif set 1-8 are misclassified to the House-X class, especially motif 1 and motif 6. It also indicates that the GNN is easily fooled by the graphs containing the plausible motifs satisfying the rule that "orange node and the red node should have 3 neighbors except the purple node, the green node and the blue node should have 4 neighbors in different color". In summary, speaking of the GNN trained on the Motif dataset with an accuracy of 0.9964, this verification study shows that our explanation results are faithful because it successfully reveals the true model pitfalls.

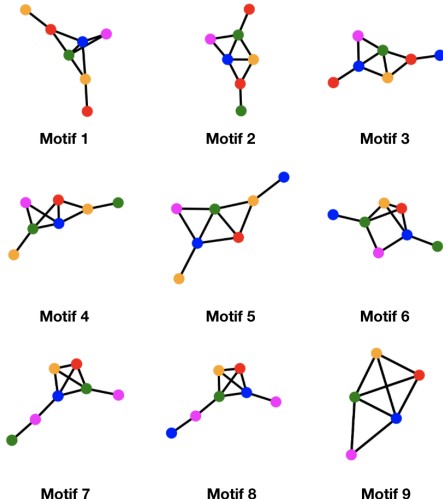

Figure 6: Plausible House-X motifs. The motifs satisfying the rule that the orange node and the red node should have 3 neighbors except the purple node, the green node and the blue node should have 4 neighbors in a different color. The motif 9 is the ground truth House-X motif. The ground truth class label of motifs 1-8 should be the Others class.

### E.2 MUTAG

According to the explanation graphs provided in Figure 1, the discriminative feature captured by our explanation graph for the mutagen class is the occurrences of the NO2 group. However, there are no special patterns observed from our explanation graph for the non-mutagen class. To verify whether the occurrences of NO2 group is truly the discriminative feature GNN tries to detect when making a prediction for the mutagen class, we designed a controlled experiment such that each set of graphs contains different numbers of NO2 group while keeping other experimental settings the same. To be specific, given 1000 randomly generated Gabriel graphs with 5-10 nodes, each time we attach 15

Table 8: The quantitative results of plausible House-X motifs. For each motif set (column), the average predicted class probability of each class (row) is computed by averaging over 5000 graphs containing the corresponding motif.

| | Motif 1 | Motif 2 | Motif 3 | Motif 4 | Motif 5 | Motif 6 | Motif 7 | Motif 8 | Motif 9 (GT) |
|---|---|---|---|---|---|---|---|---|---|
| **Others** | 0.2929 | 0.4411 | 0.3558 | 0.3849 | 0.4317 | 0.1296 | 0.3351 | 0.3373 | 0.0022 |
| **House** | 0.0000 | 0.0000 | 0.0000 | 0.0000 | 0.0000 | 0.0000 | 0.0000 | 0.0000 | 0.0000 |
| **House-X** | 0.7053 | 0.5581 | 0.6435 | 0.6144 | 0.5678 | 0.8700 | 0.6641 | 0.6620 | 0.9978 |
| **Complete-4** | 0.0002 | 0.0003 | 0.0004 | 0.0004 | 0.0004 | 0.0003 | 0.0008 | 0.0007 | 0.0000 |
| **Complete-5** | 0.0016 | 0.0004 | 0.0002 | 0.0003 | 0.0002 | 0.0001 | 0.0000 | 0.0000 | 0.0000 |

Table 9: The classification results of plausible House-X motifs. For each motif set (column), the total number of graphs that are classified as classes (row) is presented in this table. The sum of all columns should be 5000 because that is the total number of graphs in each motif set.

| | Motif 1 | Motif 2 | Motif 3 | Motif 4 | Motif 5 | Motif 6 | Motif 7 | Motif 8 | Motif 9 (GT) |
|---|---|---|---|---|---|---|---|---|---|
| **Others** | 1434 | 2251 | 1808 | 1970 | 2183 | 597 | 1682 | 1696 | 10 |
| **House** | 0 | 0 | 0 | 0 | 0 | 0 | 0 | 0 | 0 |
| **House-X** | 3566 | 2749 | 3192 | 3030 | 2817 | 4403 | 3318 | 3304 | 4990 |
| **Complete-4** | 0 | 0 | 0 | 0 | 0 | 0 | 0 | 0 | 0 |
| **Complete-5** | 0 | 0 | 0 | 0 | 0 | 0 | 0 | 0 | 0 |

nodes to these random graphs. These 15 nodes will be divided into 5 groups, and each group has 3 nodes. The frequency of NO2 group appearing in these 15 nodes varies from 0 to 5. For instance, in Figure 7, the example graph for 5 NO2 contains a Gabriel graph with 6 nodes and 5 NO2 groups; the example graph for 4 NO2 contains the same Gabriel graph, 4 NO2 groups, and a random group of 3 nodes. As a result, we will obtain 6 sets of graphs and each set includes 1000 graphs containing a fixed number of NO2 group.

Then, we feed these 6000 graphs into the GNN trained with the MUTAG dataset to obtain the quantitative results. Similar to what we did for Motif dataset, for each 1000 graphs with a fixed number of NO2 group, the average predicted class probability for each class and the number of graphs being classified as each class is presented in Table 10 and Table 11. We can clearly see that as more and more NO2 group occurs, the GNN become more and more certain about classifying these graphs as the mutagen class. That is to say, the number of occurrences of NO2 groups indeed is the discriminative features the GNN tries to detect when making a prediction for the mutagen class, which is consistent with our previous finding when analyzing our explanations for both classes in subsection 5.2. Thus, this verification study proves that our explanations for MUTAG dataset are faithful and informative because they can correctly interpret the high-level decision-making process of the GNN.

It has been shown that GNNs trained on binary classification tasks can suffer from the bias attribution issue (Faber et al., 2021). Essentially, the GNN with bias attribution issue will classify any graphs as class A unless it finds sufficient evidence of class B. From Table 10 and Table 11, it is interesting to observe that the GNN seems biased toward the non-mutagen class unless the GNN finds enough evidence (i.e., NO2 group) for the mutagen class. This indicates that this GNN is suffering from the bias attribution issue so that no special pattern is observed from our explanation graph for the non-mutagen class in Figure 1. Therefore, this verification study also proves that the faithful explanation generated by GNNInterpreter can help us to diagnose whether the GNN is suffering from the bias attribution issue.

Table 10: The quantitative results of graphs with different numbers of NO2 groups. For each 1000 graphs with the fixed number of NO2 groups (column), the average predicted class probability of each class (row) is computed by averaging over 1000 graphs containing the fixed number of NO2 group.

| 15 Nodes | 0 NO$_2$ | 1 NO$_2$ | 2 NO$_2$ | 3 NO$_2$ | 4 NO$_2$ | 5 NO$_2$ |
|---|---|---|---|---|---|---|
| **Non-Mutagen** | 0.9837 | 0.9651 | 0.9186 | 0.7593 | 0.3998 | 0.0714 |
| **Mutagen** | 0.0163 | 0.0349 | 0.0814 | 0.2407 | 0.6002 | 0.9286 |

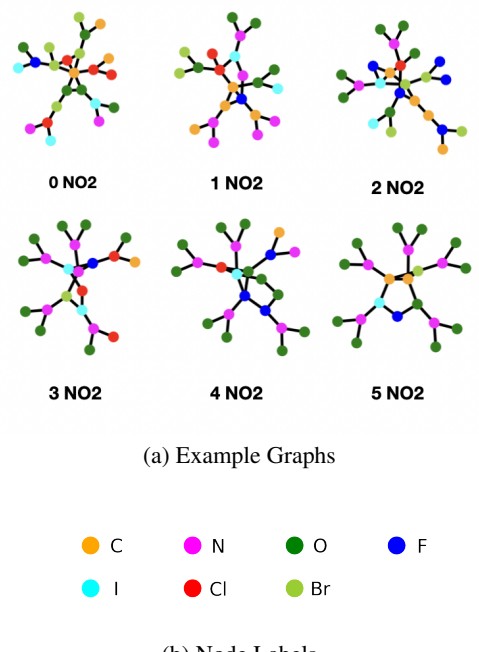

(a) Example Graphs

C ● N ● O ● F ●

I ● Cl ● Br ●

(b) Node Labels

Figure 7: The example graphs with different numbers of NO2 groups attached to the same Gabriel graph with 6 random nodes.

Table 11: The classification results of graphs with different numbers of NO2 groups. For each 1000 graphs with the fixed number of NO2 groups (column), the number of graphs being classified as the classes (row) is presented. The sum of all columns should be 1000.

| 15 Nodes | 0 NO$_2$ | 1 NO$_2$ | 2 NO$_2$ | 3 NO$_2$ | 4 NO$_2$ | 5 NO$_2$ |
|---|---|---|---|---|---|---|
| Non-Mutagen | 984 | 965 | 917 | 758 | 396 | 71 |
| Mutagen | 16 | 35 | 83 | 242 | 604 | 929 |

## F  MULTIPLE EXPLANATION GRAPHS PER CLASS PER DATASET

The redundant evidence pitfall is a potential problem in evaluating the instance-level explanation when the ground truth explanations are not unique (Faber et al., 2021). Specifically, when there are multiple ground truth explanations, comparing the generated explanation against only one ground truth explanation might yield wrong weak results. However, for a model-level explanation method, the ground truth explanation graph does not exist because we never know what the true discriminative feature the GNNs learned for each class is. Therefore, we do not rely on the ground truth explanation to evaluate our method. Instead, we can directly measure how discriminative the generated explanation is for a certain class by the corresponding class score predicted by the GNN. If the class score after softmax is close to 1, it indicates that the generated explanation indeed contains the discriminative features GNN tries to detect for a certain class. In a word, the redundant evidence pitfall in evaluating our method does not exist because we do not rely on the ground truth explanation to evaluate our method.

However, for model-level explanation methods, there might exist multiple discriminative features the GNN tries to detect for a specific class. In this case, different explanation graphs could be generated by applying our method for multiple times with different random initialization. Given that generating an explanation per class only takes less than a minute (as shown in Table 5), we believe generating multiple explanations per class is feasible in terms of the time cost. For each dataset, multiple qualitative examples per class are shown in Figure 8.

## G  EXPERIMENTAL SETUP

All the experiments are conducted on a single core of an Intel Core i9 CPU. Speaking of the software, all the models are implemented in Python 3.9. We use PyTorch 1.10 for auto-differentiation and numerical optimization. Besides, all the GNNs are implemented with PyTorch Geometric 2.0. We also use PyTorch-Scatter library to handle vector gather-scatter operations. Lastly, we utilize NetworkX for graph data generation and processing.

## H  SYNTHETIC DATASET GENERATION

In the experimental study, we evaluate the efficacy of GNNInterpreter on 4 synthetic datasets: Cyclicity, Shape, Motif, ColorConsistency, and Is_Acyclic. Among them, Cyclicity, Motif, and ColorConsistency are synthetically generated on top of Rome Dataset[2] which has 11534 undirected graphs with 10-100 nodes. The generation procedures for the first 4 synthetic datasets are specified in algorithm 3, algorithm 4, algorithm 5, and algorithm 6, respectively. Lastly, Is_Acyclic dataset is a dataset synthetically generated by the XGNN authors (Yuan et al., 2020a).

## I  EXPERIMENTAL DETAILS OF GNN CLASSIFIERS

In the experimental study, we adopt GNNInterpreter to explain 5 GNN classifiers which are trained on 5 different datasets. In this section, we will describe some experimental details about the GNN models for all 5 datasets. These experimental details include the model architecture, hyper-parameter settings, and test F1 scores per class obtained by the GNN models.

### I.1  MUTAG

The GNN classifier we implemented for MUTAG dataset is a deep GCN model that contains 3 GCN layers of width 64, a global mean pooling layer, and 2 dense layers in the end. The model uses LeakyReLU as activation. Before training, the model parameters are initialized with the Kaiming initializer. During training, the AdamW optimizer is used for optimization with a learning rate of 0.01 and weight decay of 0.01. The F1 scores for two classes in Table 12 show that the GNN is more accurate when making decisions on the Mutagen class.

---

[2]http://www.graphdrawing.org/data.html

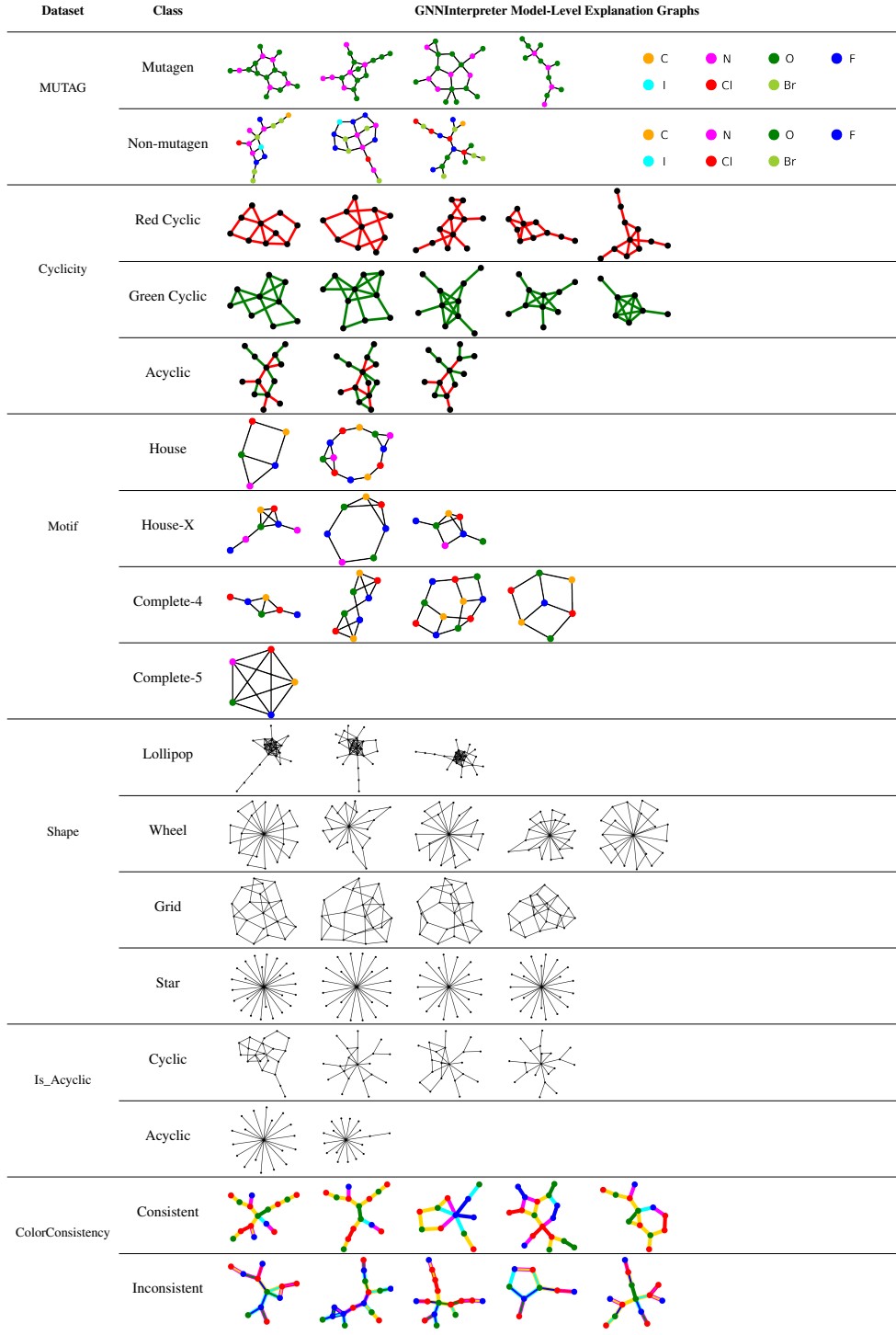

Figure 8: The qualitative results for 6 datasets. For each class in all datasets, multiple explanation graphs with the class probability of 1 predicted by the GNNs are displayed. If the dataset has the node feature or edge feature, the different colors in the nodes and edges correspondingly represent different values in the node feature and edge feature.

---

**Algorithm 3:** Cyclicity dataset generation procedure

---

**Edge Classes:** {RED, GREEN}
**Graph Classes:** {RED-CYCLIC, GREEN-CYCLIC, ACYCLIC}
**Input:** Rome Dataset
**Output:** A collection of pairs consisted of a graph and its label

1 **for** $G$ in Rome Dataset **do**
2     Randomly label each edge in $G$ as RED or GREEN.
3     **if** $G$ is acyclic **then**
4         **yield** ($G$, ACYCLIC)
5         **break**
6     **while** more than one cycle exists in $G$ **do**
7         Remove a random edge from a random cycle in $G$.
8     Randomly pick a color $C_{\text{cycle}} \in \{\text{RED, GREEN}\}$.
9     Relable all edges in the remaining cycle in $G$ as $C_{\text{cycle}}$.
10     Randomly pick a color $C_{\text{edge}} \in \{\text{RED, GREEN}\}$.
11     Relable a random edge in the remaining cycle in $G$ as $C_{\text{edge}}$.
12     **if** $C_{\text{cycle}} = C_{\text{edge}} = \text{RED}$ **then**
13         **yield** ($G$, RED-CYCLIC)
14     **else if** $C_{\text{cycle}} = C_{\text{edge}} = \text{GREEN}$ **then**
15         **yield** ($G$, GREEN-CYCLIC)
16     **else**
17         **yield** ($G$, ACYCLIC)

---

**Algorithm 4:** Motif dataset generation procedure

---

**Node Classes:** {RED, GREEN, ORANGE, BLUE, MAGENTA}
**Graph Classes:** {HOUSE, HOUSE-X, COMPLETE-4, COMPLETE-5, OTHERS}
**Input:** Rome Dataset
**Output:** A collection of pairs consisting of a graph and its label

1 Let $\{G_{\text{HOUSE}}, G_{\text{HOUSE-X}}, G_{\text{COMPLETE-4}}, G_{\text{COMPLETE-5}}\}$ be 4 motifs for the corresponding classes.
2 Define $\bigoplus$ as a graph operator such that $G_1 \bigoplus G_2$ generates a union graph $G_1 \cup G_2$ with an additional edge between a random node in $G_1$ and a random node in $G_2$.
3 **for** $G$ in Rome Dataset **do**
4     Randomly assign a color $C_{\text{node}} \in \{\text{RED, GREEN, ORANGE, BLUE, MAGENTA}\}$ for each node in $G$
5     Randomly select a label $L \in \{\text{OTHERS, HOUSE, HOUSE-X, COMPLETE-4, COMPLETE-5}\}$.
6     **if** $L = \text{OTHERS}$ **then**
7         Let $G_{\text{OTHERS}}$ be a random motif in $\{G_{\text{HOUSE}}, G_{\text{HOUSE-X}}, G_{\text{COMPLETE-4}}, G_{\text{COMPLETE-5}}\}$ but with a random edge being removed.
8     **yield** ($G \bigoplus G_L$, $L$)

---

## I.2 CYCLICITY

The GNN classifier for Cyclicity dataset is a deep NNConv model that contains 5 NNConv layers of width 32 with a single-layered edge network, a global mean pooling layer, and 2 dense layers in the end. Before training, the model parameters are initialized with Kaiming initializer. During training, AdamW optimizer is used for optimization with a learning rate of 0.01, a learning rate decay of 0.01, and a weight decay of 0.01. As shown in Table 12, all three classes have similar F1 scores, which means that the GNN model has a balanced performance on each class.

## I.3 MOTIF

The Motif dataset uses exactly the same model architecture and hyper-parameter settings as the MUTAG dataset. The F1 score per class presented in Table 12 indicates that the GNN achieves a perfect performance on Complete-5 class and a near-perfect performance on every other class.

## I.4 SHAPE

The Shape dataset uses the same model architecture and hyper-parameter settings as the MUTAG dataset except that the model contains 4 GCN layers instead of 3 GCN layers. Also, the F1 score per

---

**Algorithm 5:** Shape dataset generation procedure

---

**Graph Classes:** {LOLLIPOP, WHEEL, GRID, STAR, OTHERS}
**Output:** A collection of pairs consisted of a graph and its label

1 **for** *8000* times **do**
2     Randomly select a label $L \in$ {OTHERS, LOLLIPOP, WHEEL, GRID, STAR}.
3     **switch** $L$ **do**
4         **case** LOLLIPOP **do**
5             Sample lollipop graph $G_{\text{LOLLIPOP}}$ with random number of head nodes $n \in \{4, ..., 16\}$ and random number of tail nodes $m \in \{4, ..., 16\}$.
6         **case** WHEEL **do**
7             Sample wheel graph $G_{\text{WHEEL}}$ with random number of non-center nodes $n \in \{4, ..., 64\}$.
8         **case** GRID **do**
9             Sample grid graph $G_{\text{GRID}}$ with random width $w \in \{2, ..., 8\}$ and random height $h \in \{2, ..., 8\}$.
10         **case** STAR **do**
11             Sample star graph $G_{\text{STAR}}$ with random number of non-center nodes $n \in \{4, ..., 64\}$.
12         **case** OTHERS **do**
13             Sample Binomial random graph $G_{\text{OTHERS}}$ with random number of nodes $n \in \{8, ..., 32\}$ and random edge probability $p \in [0.2, 1]$.
14     Add random number of noisy edges to $G_L$ with a random ratio $p \in [0, 0.2]$.
15     **yield** $(G_L, L)$

---

**Algorithm 6:** ColorConsistency dataset generation procedure

---

**Node Classes:** {RED, GREEN, BLUE}
**Edge Classes:** {RED, GREEN, BLUE, YELLOW, CYAN, MAGENTA}
**Graph Classes:** {CONSISTENT, INCONSISTENT}
**Output:** A collection of pairs consisting of a graph and its label

1 **for** $G$ in Rome Dataset **do**
2     Randomly assign a color $C_v \in$ {RED, GREEN, BLUE} for each node $v$ in $G$
3     **for** each edge $e = (u, v)$ **do**
4         Assign color $C_e = C_u \mid C_v \in$ {RED, GREEN, BLUE, YELLOW, CYAN, MAGENTA},
5         where $\mid$ denote an operator to mix two colors.
6     Randomly select a label $L \in$ {CONSISTENT, INCONSISTENT}.
7     **if** $L$ = INCONSISTENT **then**
8         **for** each $e$ in $K$ random edges of $G$, where $K <$ # of edges in $G$ **do**
9             Reassign a random color $C'_e \in$ {RED, GREEN, BLUE, YELLOW, CYAN, MAGENTA}$\setminus\{C_e\}$
10     **yield** $(G, L)$

---

class in Table 12 shows that the GNN can perfectly predict the Star class, whereas the predictions for the Lollipop class and the Grid class are less accurate.

### I.5 COLORCONSISTENCY

The ColorConsistency dataset has both node features and edge features. So, we take advantage of the GAT model Veličković et al. (2018) to classify this type of graph data. The GAT model we use contains 3 GAT layers of width 16, a global sum pooling layer, and 2 dense layers in the end. Before training, the model parameters are initialized with the Kaiming initializer. During training, the AdamW optimizer is used for optimization with learning rate of 0.001, a learning rate decay of 0.01, and a weight decay of 0.01. The model is able to predict both classes with F1 scores very close to 1.0.

### I.6 IS_ACYCLIC

The Is_Acyclic dataset uses the same model architecture and hyper-parameter settings as the Shape dataset except that max pooling is used instead of mean pooling for the global pooling layer, due to the nature of the task that the class prediction can be very sensitive to local patterns. The model is able to predict both classes perfectly with an F1 score of 1.0.

Table 12: The F1 score per class of the corresponding GNN classifiers for all datasets.

| Dataset | F1 Scores of GNN Classifier | | | |
|---|---|---|---|---|
| MUTAG | Mutagen 0.961 | Nonmutagen 0.917 | | |
| Cyclicity | Red Cyclic 0.995 | Green Cyclic 0.992 | Acyclic 0.993 | |
| Motif | House 0.998 | House-X 0.998 | Complete-4 0.996 | Complete-5 1.000 |
| Shape | Lollipop 0.969 | Wheel 0.994 | Grid 0.966 | Star 1.000 |
| Is_Acyclic | Cyclic 1.000 | Acyclic 1.000 | | |
| ColorConsistency | Consistent 0.9963 | Inconsistent 0.9962 | | |

Table 13: The regularization weights of GNNInterpreter for explaining the GNN models corresponding to each dataset.

| Dataset | Class | Regularization Weights | | | |
|---|---|---|---|---|---|
| | | $R_{L_1}$ | $R_{L_2}$ | $R_b$ | $R_c$ |
| MUTAG | Mutagen | 10 | 5 | 20 | 1 |
| | Nonmutagen | 5 | 2 | 10 | 2 |
| Cyclicity | Red Cyclic | 10 | 5 | 10000 | 100 |
| | Green Cyclic | 10 | 5 | 2000 | 50 |
| | Acyclic | 10 | 2 | 5000 | 50 |
| Motif | House | 1 | 1 | 5000 | 0 |
| | House-X | 5 | 2 | 2000 | 0 |
| | Complete-4 | 10 | 5 | 10000 | 1 |
| | Complete-5 | 10 | 5 | 10000 | 5 |
| Shape | Lollipop | 5 | 5 | 1 | 5 |
| | Wheel | 10 | 5 | 10 | 0 |
| | Grid | 1 | 1 | 2 | 0 |
| | Star | 10 | 2 | 200 | 0 |
| Is_Acyclic | Cyclic | 0 | 0 | 500 | 0 |
| | Acyclic | 1 | 5 | 500 | 5 |
| ColorConsistency | Consistent | 1 | 1 | 50 | 1 |
| | Inconsistent | 0 | 0 | 50 | 1 |

## J   EXPERIMENTAL DETAILS OF GNNINTERPRETER

For all experiments, we choose to set the Concrete Distribution temperature $\tau = 0.2$. We use the sample size $K = 10$ for all Monte Carlo samplings. For optimization, we adopt the SGD optimizer with a learning rate of 1 and terminate the learning procedure until convergence. Additionally, in the training objective, the embedding similarity weight $\mu$ is set to 1 for every dataset except MUTAG. For the MUTAG dataset, we let $\mu = 10$ due to its nature that the structure of the graphs in the dataset needs to follow certain domain-specific rules. A higher value of $\mu$ allows the GNNInterpreter to implicitly capture these rules from the average graph embeddings. Speaking of the regularization weight, we employ grid search to obtain the best results under different combinations of regularization terms. To ensure the reproducibility of our experimental results, we also report the exact regularization weights used for all the quantitative and qualitative evaluations from the manuscript in Table 13. Note that we adopt a constant weight strategy for L1 regularization $R_{L_1}$, L2 regularization $R_{L_2}$, and connectivity constraint $R_c$ throughout the training process. In contrast, for budget penalty $R_b$, we employ a 500-iteration initial warm-up period to gradually increase the weight starting from 0.

