# OpenReview forum: "GNNInterpreter: A Probabilistic Generative Model-Level Explanation for Graph Neural Networks"
_ICLR.cc/2023/Conference — ICLR 2023 poster_

### Official Review · Reviewer_B5NJ · 2022-10-21

**Confidence:** 4
**Correctness:** 4
**Technical Novelty And Significance:** 3
**Empirical Novelty And Significance:** Not applicable
**Recommendation:** 8

**Clarity, Quality, Novelty And Reproducibility:**

Novelty:
Although generating a graph to explain the GNN model is not a brand new idea, this approach is still novel and it provides a significant advance to the field.

Quality:
The proposed method is well supported both theoretically and empirically, especially when combined with the appendix. The assumptions, though not very strong, might need justification.

Clarity:
The method is clearly described and discussed.

Reproducibility:
The experimental details are provided in the appendix. The code has not been provided yet at the time of the review.

**Strength And Weaknesses:**

Strength:
1. Great ambition in attempting to interpret GNN models on different types of graphs.
2. By using an objective function instead of adopting the reinforcement learning setting, the efficiency is greatly improved.
3. The relaxation on the adjacency matrix and the features is clearly described.

Weakness:
1. Need to clarify that this method focuses on graph classification task, similar to XGNN.
2. None of the experiments is carried out on a graph with both node features and edge features.
3. Due to the space limitation, the algorithms and most of the figures are in the appendix, making the main body of the paper a little bit dry.
4. In "The lack of self-explainability becomes a serious obstacle for applying GNNs to real-world problems, especially those when making wrong decisions may incur an unaffordable cost" the "those" should be removed.

Suggestion:
1. Try to move Algorithm 1 back into the main body of the paper. It's essential for understanding the whole structure of GNNInterpreter.
2. Instead of continuous relaxation, maybe optimization methods on the discrete settings could be considered as alternatives, such as subgradient.


**Summary Of The Paper:**

This paper presents a probabilistic generative model-level explanation method for GNN named GNNInterpreter. By relaxing the adjacency matrix, the node features, and the edge features to continuous variables, it generates a graph capturing the critical characteristics that the GNN models discriminate the classes.

Compared to its counterpart XGNN, GNNInterpreter is capable of generating graphs with continuous node features as well as edge features. It also requires no domain expertise or knowledge.

**Summary Of The Review:**

The interpreter for GNN presented in the paper outperforms XGNN, which is the SOTA GNN model-level explainer. Most of the paper is well-written and the proposed method is well-supported. Some minor weaknesses need to be fixed.

---

> ### Author Response · Authors · 2022-11-19
> **Response to Reviewer B5NJ**
>
> We sincerely appreciate your valuable time and efforts in reviewing this paper. We thank the thoughtful feedbacks you provided, which significantly improved the quality of this paper. For the potential concerns you bring up, we would like to answer/address them as the followings.
>
> >Need to clarify that this method focuses on graph classification task, similar to XGNN.
>
> Thanks for bringing up this lack of clarity. **We have revised the introduction section and the related work section to clarify that.**
>
> >None of the experiments is carried out on a graph with both node features and edge features.
>
> Thanks for your valuable comment. We totally agree that the experiment on a graph with both node features and edge features is crucial to demonstrate the generalizability of our method. We are more than happy to adopt your valuable suggestion and conduct an experimental study on generating explanation graphs with both node features and edge features for a GAT model. The results show that we can generate faithful explanations with both node features and edge features. **Please refer to Appendix D.3 for detailed evaluation results on explaining a graph classification task with both node features and edge features.**
>
> >Due to the space limitation, the algorithms and most of the figures are in the appendix, making the main body of the paper a little bit dry. Try to move Algorithm 1 back into the main body of the paper. It's essential for understanding the whole structure of GNNInterpreter.
>
> Thanks for your constructive comment. It is very unfortunate that the page limit does not allow us to put more experimental details, more experimental results, and more details about our methodology in the main body. We also agree that Algorithm 1 is very important for obtaining a high-level understanding of our method. **Therefore, we decided to put Algorithm 1 in the main text, and also describe more about the regularization term in the main text.**
>
> >In "The lack of self-explainability becomes a serious obstacle for applying GNNs to real-world problems, especially those when making wrong decisions may incur an unaffordable cost" the "those" should be removed.
>
> Thanks for pointing this out. **We have deleted “those” in the manuscript.**
>
> >Instead of continuous relaxation, maybe optimization methods on the discrete settings could be considered as alternatives, such as subgradient.
>
> We totally agree that the subgradient could be a good alternative to continuous relaxation. Thanks for your thoughtful feedback. It lights up an interesting direction for our future research on explaining GNNs.
>
> >The assumptions, though not very strong, might need justification.
>
> Thanks for bringing up this lack of justification. **We provide a detailed justification of two assumptions in Appendix C for your reference.**
>
> ### Please kindly let us know if you have any concerns you find not fully addressed. We are more than happy to have a further discussion regarding it.

---

> > ### Comment · Reviewer_B5NJ · 2022-11-25
> > **Response to the authors**
> >
> > I appreciate your efforts in revising the paper and solving my concerns. I raised the rate from 6 to 8.

---

### Official Review · Reviewer_JE8w · 2022-10-21

**Confidence:** 5
**Correctness:** 3
**Technical Novelty And Significance:** 3
**Empirical Novelty And Significance:** 3
**Recommendation:** 6

**Clarity, Quality, Novelty And Reproducibility:**

This paper is clearly written, while some repetitive sentences need to be avoided. The authors proposed a novel method, probabilistic generative model-level explanation, for GNN considering features of nodes and edges. Experiments are evaluated on both synthetic and real-world datasets. However, there is no code in the supplementary materials.

**Strength And Weaknesses:**

**Strengths:**
1. It is novel that this paper learns a probabilistic generative graph distribution to generate an explanation graph with different types of node features and edge features.
2. The experimental results are analyzed in detail, and the proposed method is tested on several datasets.

**Weaknesses:**
1. Some of the latest related works on the instance-level explanation of GNN are not thoroughly discussed, such as causal explanation GEM [1] and OrphicX [2], reinforcement learning explanation RG-Explainer [3] and perturbation-based explanation SubgraphX [4].
2. The experiments in this paper to evaluate explanation only focus on graph classification, while other GNNs tasks are also crucial in explainability research, such as node classification and link prediction.
3. In Section 4.2, the authors present two assumptions to learn a probabilistic generative graph distribution, but an explanation of the reasonableness of the assumptions is missing.
4. In Section 4.3, the reparameterization trick is efficient for the sampling function to be differentiable. However, it is not new to use this idea which has been used by previous work such as PGExplainer [5].
5. It is said that "instance-level methods are very time-consuming," but the authors do not compare the proposed method with these instance-level models to illustrate this claim. It is suggested to add relevant comparative experiments to support this claim.
6. In addition, it does not analyze the time complexity and shows why GNNInterpreter saves more time than XGNN, which is critical for showing the effectiveness of the proposed method.
7. In Section 5.2, the experiments only choose GCN or NNConv as GNN models, which are not comprehensive to illustrate model-agnostic on other GNN models such as GAT and GraphSage.
8. In Section 5.2, the experiments in the verification study show the reasonableness of speculation to House-X class. However, it does not adequately explain the reasons for the difference between the explanations and ground truth in House-X and Complete-4.

[1] Generative Causal Explanations for Graph Neural Networks. ICML, 2021.
[2] OrphicX: A Causality-Inspired Latent Variable Model for Interpreting Graph Neural Networks. CVPR, 2022.
[3] Reinforcement Learning Enhanced Explainer for Graph Neural Networks. NeurIPS, 2021.
[4] On Explainability of Graph Neural Networks via Subgraph Explorations. ICML, 2021.
[5] Parameterized Explainer for Graph Neural Network. NeurIPS, 2020.



**Summary Of The Paper:**

This paper proposes GNNInterpreter to detect a model-level explanation for GNN. GNNInterpreter learns a probabilistic generative graph distribution to detect the model-level explanation without introducing another model. Experiments are evaluated on both synthetic and real-world datasets.

**Summary Of The Review:**

This paper is easy to read and proposes a new approach to explain GNN models with a probabilistic generative graph. The author should explain the problem, followed by laying out their arguments more clearly. More experiments enforcing the arguments empirically are needed. Overall, the current version of this paper is not ready to be published in ICLR.

---

> ### Author Response · Authors · 2022-11-19
> **Response to Reviewer JE8w [5/5]**
>
> >This paper is clearly written, while some repetitive sentences need to be avoided.
>
> Thanks for your kind words. **We have revised the manuscript to avoid repeating the same sentences multiple times.**
>
> >The author should explain the problem, followed by laying out their arguments more clearly.
>
> Thanks for your valuable suggestions. **We have revised the introduction section of our paper to explain our problem and argument more clearly.**
>
> ### Please kindly let us know if you have any concerns you find not fully addressed. We are more than happy to have a further discussion regarding it.

---

> > ### Comment · Reviewer_JE8w · 2022-11-26
> > **Response to the Authors**
> >
> > Thanks to the authors for their detailed responses to my concerns. I think the questions in the review have been answered, which has led me to raise the score.

---

> ### Author Response · Authors · 2022-11-19
> **Response to Reviewer JE8w [4/5]**
>
> >In Section 5.2, the experiments only choose GCN or NNConv ... other GNN models such as GAT and GraphSage.
>
> Thanks for your helpful suggestion. It is true that evaluating our method on GCN and NNConv might not be comprehensive enough. GAT is a very representative graph convolutional layer because it takes advantage of the attention mechanism, which is different from GCN and NNConv to a large extent. We are more than happy to adopt your suggestion and conduct an experimental study to evaluate our method on GAT. To further demonstrate the flexibility of our explanation method, this experimental study is conducted on a synthetic dataset with both node features and edge features. The result shows that our method can generate a faithful explanation graph with both node features and edge features for that GAT model. **Please refer to Appendix D.3 for detailed evaluation results on explaining the GAT classifier.**
>
> Speaking of evaluating our method on GraphSage, it is a very good point because GraphSage is also a widely used graph convolutional layer. However, as mentioned in the GraphSage paper [1], Graphsage is just a generalized version of GCN, in terms of the message aggregation function. Since GraphSage is concerned more about learning from large graphs, GraphSAGE can be regarded as a GCN with subsampled neighbors. We believe that our evaluation results on GCN can sufficiently demonstrate the effectiveness of our method on GraphSage. Lastly, we hope we can empirically evaluate our method on every existing GNN model, but that might not be a realistic goal because, as far as we know, there are about 60 graph convolutional layers have been implemented in Pytorch Geometric. Theoretically speaking, our method can explain different GNNs following the message passing scheme for the reason mentioned in Section 4.4.
>
> >In Section 5.2, ... it does not adequately explain the reasons for the difference between the explanations and ground truth in House-X and Complete-4.
>
> Thanks for bringing up this lack of clarity. The difference between the explanation and ground truth in House-X and Complete-4 indicates that the GNN model might be “cheating.” Again, take the image classifier of Apples and Bananas as an example. Suppose this classifier learns to use color to differentiate apples and bananas. The discriminative features this classifier tries to detect for the apple class and the banana class are red color and yellow color, respectively. Then, an image of pure red color might be predicted as the Apple with a class probability close to 1 because it is full of the discriminative feature of the Apple class. We could expect that the model-level explanation image for the Apple class can be a pure red image. However, ideally, we hope the classifier can use both the object shape and color pattern to classify Apples and Bananas. Therefore, the difference between explanation and ground truth actually indicates that the model might be cheating by using some partial discriminative features as evidence to make decisions.
>
> Take the Complete-4 class in our paper as an example. The explanation graph for Complete-4 we present in Figure 1 is predicted as the Complete-4 class by GNN with the class probability of 1. Thus, our explanation graph should contain the discriminative features of Complete-4, otherwise, the GNN would not predict it as Complete-4 with a class probability of 1. If we analyze the explanation graph of Complete-4, we can see that it follows the rule that “every node should have 3 neighbors in different colors”. With this rule, it is sufficient for the GNN model to differentiate Complete-4 from the other 3 classes. However, the ground-truth discriminative features of the Complete-4 class is “in a motif with 4 nodes, every node should have 3 neighbors in different colors”. Therefore, the discriminative feature of the Complete-4 class learned by the GNN is just a partial feature. **To verify this is the true features of the Complete-4 class perceived by the GNN model, we also present a verification study for the Complete-4 class in Appendix D.**
>
> In summary, if our explanation graph matches the expected ground-truth discriminative features, this indicates that the GNN model is perfect. However, if our explanation graph does not match the expected ground-truth discriminative features, this implies that the GNN model might make mistakes for some unseen graphs with misleading patterns ( a partly correct discriminative pattern). That is why our method is useful for examining the model reliability because we can help to identify some potential model pitfalls (e.g., bias attribution issue, misclassification of unseen graphs with the misleading pattern, etc.). **We have revised Section 5.2 to make the reasons for the difference between the explanation graph and the ground truth to be more clear.**
>
> ## Reference
> [1] William Hamilton, Rex Ying, and Jure Leskovec. Inductive representation learning on large graphs. 06 2017.

---

> ### Author Response · Authors · 2022-11-19
> **Response to Reviewer JE8w [3/5]**
>
> >It is said that "instance-level methods are very time-consuming," but the authors do not compare the proposed method with these instance-level models to illustrate this claim. It is suggested to add relevant comparative experiments to support this claim.
>
> Thanks for bringing up this lack of clarity. In the paper, we said that “If the ultimate goal is to examine the model reliability, instance-level methods are very time-consuming because one will need to examine many instances one by one to draw a rigorous conclusion about the model reliability.” The reason behind this statement is that the model-level explanation method has a completely different goal than the instance-level explanation method. Basically, the model-level explanation method is designed to interpret the high-level decision-making process directly, while the instance-level explanation method is designed to explain why a certain prediction is made for a specific graph instance. Therefore, if the goal is to examine the model reliability, the instance-level explanation method might not be a good choice because we cannot draw a rigorous conclusion regarding the high-level decision-making process only based on the instance-level explanation of a single graph instance. We might need to run the instance-level explanation method for different instances and then analyze their corresponding explanations one by one in order to examine the model reliability. However, for the model-level explanation method, you just need to analyze a single generated explanation graph in order to examine the model reliability. **If the discriminative features of this class deviate from the true discriminative features, it indicates a potential risk of misclassification of unseen graphs with misleading patterns, as shown in our verification study (see Section 5.2 and Appendix E).**
>
> That is the reason why we said that “if the ultimate goal is to examine the model reliability, instance-level methods are very time-consuming”. The “time-consuming” is not talking about the computational complexity of the instance-level explanation method. Instead, it is the progress of using the instance-level explanation method to check the high-level model reliability that is “time-consuming”. Therefore, we regret that we are unable to conduct comparative experiments to support this cognitive claim. Besides, we do realize that the wording of this sentence is misleading. **We have revised this sentence in the manuscript as: “If the ultimate goal is to examine the model reliability, one will need to examine many instance-level explanations one by one to draw a rigorous conclusion about the model reliability, which is cumbersome and time-consuming.”**
>
> In addition, the explanations generated by our method are more informative about examining the model reliability than the instance-level explanation graphs. Theoretically, we can categorize all the truly existing graphs into three groups: seen graphs (train set), known but unseen graphs (test set), unknown and unseen graphs. Specifically, the third category contains all the truly existing graph which has not been collected/observed yet. The instance-level explanation can only explain the decision-making process of observed graphs, while our method can provide insights into how GNN would behave on graphs that have not been observed yet. In this work, the explanation graphs are synthetically generated to maximize the corresponding class score, which might not be observed yet. If the ground truth label of our synthetically generated explanation graph $EG_{c1}$ for class c1 is c2, it actually conveys a message that the model is not reliable for classifying unobserved graphs of class c2 with similar features as $EG_{c1}$.
>
> >It does not analyze the time complexity and shows why GNNInterpreter saves more time than XGNN, which is critical for showing the effectiveness of the proposed method.
>
> Thanks for pointing this out. We strongly agree that the computational time computed empirically might not be sufficient to evaluate the efficiency of our method. **Therefore, we conducted a detailed time complexity analysis for our method and XGNN in Appendix B.** Our time complexity is $O(TKN^2)$, where $T$ is the number of iterations, $K$ is the number of Monte-Carlo samples, $N$ is the number of vertices for the Gilbert random graph distribution. The time complexity of XGNN is $O(TRM^3)$, where $T$ is the number of episodes, $R$ is the number of Rollouts, and $M$ is the maximum number of edges in the explanation graph. For a connected explanation graph with $N$ nodes and $M$ edges, $M > N-1$ always holds. Therefore, $O(TRM^3)$ is much higher than $O(TKN^2)$ since $R$ and $K$ are constant hyper-parameters. Our method outperforms XGNN in terms of time complexity. Besides, according to the quantitative evaluation result in Table 5, it only takes less than a minute for our method to explain one class in a GNN model, which is as many as 10 times faster than XGNN.

---

> ### Author Response · Authors · 2022-11-19
> **Response to Reviewer JE8w [2/5]**
>
> >The experiments in this paper to evaluate explanation only focus on graph classification, while other GNNs tasks are also crucial in explainability research, such as node classification and link prediction.
>
> We also believe that explaining the node classification task and the link prediction task is very crucial. However, developing a model-level explanation method for other machine learning tasks on graph data might deserve writing another paper. The reason is that the problem formulation of model-level explanation is very task-dependent, unlike instance-level explanation methods. If we want to develop a model-level explanation method for node classification and link prediction, we first need to carefully define what the model-level explanation for those tasks is and then design an algorithm to generate the explanations. Therefore, this is a nontrivial task that requires extensive effort. For the same reason, all the existing model-level explanation methods for GNNs also only focus on explaining graph classification task. We sincerely appreciate your constructive comment. It lights up an interesting and important research direction for model-level explanation methods of GNNs.
>
> >In Section 4.2, the authors present two assumptions to learn a probabilistic generative graph distribution, but an explanation of the reasonableness of the assumptions is missing.
>
> Thanks for your insightful comment. **The reasonableness of our assumption is described in Appendix C for your reference.**
>
> >In Section 4.3, the reparameterization trick is efficient for the sampling function to be differentiable. However, it is not new to use this idea which has been used by previous work such as PGExplainer [5].
>
> Thanks for your thoughtful comment. We totally agree that adopting the Gumbel-Softmax trick in GNN literature is not a new idea. We are the first method that adopts the Gumbel-Softmax trick (GS trick) to generate model-level explanations for GNN, but we are not the first method adopting the GS trick in GNN literature. Therefore, we do not consider adopting the GS trick as our core innovation. We would rather regard the reparameterization trick as part of our methodology because it has become a standard practice in probabilistic machine learning.
>
> We would like to take this opportunity to emphasize the core innovation of this work. Our novelty is mainly about designing a new objective function for generating a meaningful model-level explanation for GNNs. Specifically, the special design of our loss function allows us to generate a more realistic explanation graph with the discriminative and representative features of that class perceived by the GNN model. The activation maximization term (the first term from equation 2) enforces the explanation graph to capture the most discriminative features of that class perceived by the GNNs. The cosine similarity term (the second term from equation 2) encourages the explanation graph to capture the representative features of that class because the mean embedding computed over the training data actually encapsulates the representative features of this class perceived by the GNN model itself. Compared with the state-of-the-art model-level explanation method for GNNs, we are more computationally efficient, more flexible in explaining GNNs in different application domains without requiring manually defined domain-specific rules, and more applicable to broader usage scenarios with different types of node features and edge features. We believe that this work has some unique value in diagnosing and identifying the potential pitfalls of GNNs.

---

> ### Author Response · Authors · 2022-11-19
> **Response to Reviewer JE8w [1/5]**
>
> We sincerely appreciate your valuable time and efforts in reviewing this paper. We thank the thoughtful feedbacks you provided, which significantly improved the quality of this paper. For the potential concerns you bring up, we would like to answer/address them as the followings.
>
> >Some of the latest related works on the instance-level explanation of GNN are not thoroughly discussed, such as causal explanation GEM [1] and OrphicX [2], reinforcement learning explanation RG-Explainer [3] and perturbation-based explanation SubgraphX [4].
>
> Thanks for your kind suggestions. We strongly agree that we have missed some important references in our paper, including several most recent instance-level explanation methods. **We have revised our related work section accordingly, which briefly summarizes these methods (e.g., GEM, OrphicX, RG-Explainer, and SubgraphX).** Due to the page limit, we regret that we are unable to discuss every instance-level explanation method in detail because most instance-level methods, in fact, are not very closely related to this paper.
>
> Instance-level methods for GNNs are very useful in explaining why a certain prediction is made for a specific instance. They can identify the instance-level evidence which supports GNNs’ prediction. They are very good at explaining the GNNs’ decisions but might not be suitable for interpreting the high-level decision-making processes of GNNs. That’s why we need the model-level explanation for GNNs, which are still much less explored than the instance-level explanation for GNNs. To be specific, the model-level explanation methods for GNNs attempt to identify the discriminative features GNNs try to detect for a certain class. The discriminative features of class C are encapsulated into an explanation graph which maximizes the score of class C. For example, we train an image classifier to distinguish between apples and bananas. Suppose, this classifier learns to use color to differentiate apples and bananas. The discriminative features this classifier tries to detect for the apple class and the banana class are red and yellow, respectively. Then, we could expect that the model-level explanation for the apple class is just a pure red image, whereas the model-level explanation for the banana class is just a pure yellow image. Based on this, the potential model pitfall of this classifier is that it will classify any image containing red color to be the apple class. In summary, the ultimate goal of the model-level explanation method is to generate a global explanation for each class such that the class-specific explanation graph can be used to examine the model reliability. As shown in our verification study (see Section 5.2 and Appendix E), our method can be used to identify the bias-attribution issue, the potential risk of misclassification of unseen graphs with misleading patterns, etc. Therefore, our method actually has a completely different goal than the instance-level explanation methods for GNNs.
>
> However, we do realize that some instance-level explanation methods are worthy of being thoroughly discussed in our paper, even though we have different goals. Specifically, PGExplainer is closely related to our work because they also adopt the Gumbel-Softmax trick to learn the explanation distribution. The main difference between PGExplainer and our method is that we learn a model-level explanation distribution obtained by gradient ascent, but they learn an instance-level explanation distribution predicted by an MLP. Besides, our objective function is maximizing the activation and the cosine similarity between the explanation embedding and mean embedding, while their objective function is maximizing the mutual information between the GNN prediction of the explanation subgraph and the GNN prediction of the original graph. **We have revised the related work section to discuss our relationship with PGExplainer thoroughly.**

---

### Official Review · Reviewer_jhWR · 2022-10-21

**Confidence:** 5
**Correctness:** 4
**Technical Novelty And Significance:** 2
**Empirical Novelty And Significance:** 2
**Recommendation:** 8

**Clarity, Quality, Novelty And Reproducibility:**

The paper is well written and easy to follow. The idea is clear from first reading. There are many important elements of the method (including several regularization terms) that are relegated to the appendix. In my opinion it would be better to move these to the main text, since they are essential for reproducing the results.

**Strength And Weaknesses:**

First, the method which is proposed is basically activation maximization applied to a graph, with a simple "localization" term (see, e.g., [1]) to force the embeddings to lie close to the embeddings of the real graphs. I would suggest to discuss explicitly the relation of the method with known methods from the CV literature. In my experience, in the CV literature this kind of localization regularization is uncommon because it tends to "collapse" the answer on the reference point (the average of the embeddings). It would be interesting for the authors to evaluate whether this is also happening here (i.e., what happens if I simply take the $\hat{\phi}$ term from (2) and visualize it after proper thresholding?).

The core innovation of the work is using the GS trick to reparameterize the gradients. Also here, this is quite common today in the GNN literature (e.g., DGM for latent graph imputation, PGExplainer for explanation, adaptive graph sparsification, ...). There are also many recent proposals to generate discrete structures that do not require to assume independence of all edges, which is really strong [2]. I believe it would be better to discuss these applications on the paper, as they are clearly related (especially PGExplainer and similar works).

Third, the authors mention several time that "the generated explanation is both faithful and valid regarding the domain-specific knowledge", but this cannot guaranteed since this is only enforced through a weak regularization (indirectly via the embeddings). Unless this is tested more extensively, I believe these sentences are not valid.

[1] https://www.sciencedirect.com/science/article/pii/S1051200417302385
[2] https://arxiv.org/abs/2106.01798

**Summary Of The Paper:**

The paper proposes a model-level explanation method for GNNs. The core idea is, given a class, to maximize a graph that has the highest softmax-activation for that class. In order to obtain graphs that are meaningful, a regularization term is added forcing similarity of the graph embedding with the average embeddings for the same class. Because many components of a graph are discrete (e.g., the presence or absence of an edge), they apply the Gumbel-Softmax trick to reparameterize the gradients, under the assumption that all edges can be independently sampled. They compare the method with a baseline method, XGNN, or a series of standard benchmarks.

**Summary Of The Review:**

Overall, I believe the results are interesting from the point of view of xAI on graphs. However, the method is a simple adaptation of techniques from the CV world. The experiments lack some important points to showcase the validity of the methods. Because of this, I think this is at most a borderline paper in terms of the ICLR audience, that could benefit from the points raised above, especially concerning a more proper evaluation of the state-of-the-art and more experimental details.

---

> ### Author Response · Authors · 2022-11-19
> **Response to Reviewer jhWR [3/3]**
>
> >There are also many recent proposals to generate discrete structures that do not require to assume independence of all edges, which is really strong [2].
>
> Thanks for bringing this up. We totally agree that I-MLE[6] is a good alternative to the Gumbel-Softmax trick (GS trick), especially considering that I-MLE does not require the independence assumption. But the effectiveness of I-MLE has been less tested by the community so far. In contrast, GS trick is more well-established and has become a standard practice in probabilistic machine learning. Besides, as shown in Figure 4 from [6], the experimental result of a discrete k-subset Variational Auto-Encoder shows that GS trick and I-MLE achieve comparable results in terms of the reconstruction loss value. For all the reasons above, we believe the GS trick is a reasonable choice for this work.
>
> Speaking of the independence assumption we have made, our explanation method, in fact, takes the dependence between edges into account when learning the explanation graph distribution. The reason is that the training process will compute the gradient through the message-passing operation inside GNN, and the message-passing operation exchanges information across the whole graph in the forward pass. Therefore, the gradient computed backward will capture the correlation among latent parameters of the independent distributions. Through this mechanism, the dependency among edge distributions, edge feature distributions, and node feature distributions are all taken into account in the learning process. That is to say, the independence assumption we made is only applied to sampling an explanation graph from the learned explanation graph distribution, while the latent parameters of those distributions are actually correlated under the hood. **The detailed justification of the two assumptions we made is presented in Appendix C for your reference.**
>
>
> >I believe it would be better to discuss these applications on the paper, as they are clearly related (especially PGExplainer and similar works).
>
> Thanks for your valuable suggestions. **We have revised our related work section accordingly to discuss PGExplainer in detail.**
>
> >Third, the authors mention several times that "the generated explanation is both faithful and valid regarding the domain-specific knowledge,” but this cannot be guaranted since this is only enforced through a weak regularization (indirectly via the embeddings). Unless this is tested more extensively, I believe these sentences are not valid.
>
> Thanks for your kind suggestions. After reading your comment, we also realized some sentences like the one you quoted here are not rigorous enough. **We have revised the manuscript accordingly with more appropriate wording.**
>
> ## Reference
> [1]Grégoire Montavon, Wojciech Samek, Klaus-Robert Müller, Methods for interpreting and understanding deep neural networks, Digital Signal Processing, Volume 73, 2018, Pages 1-15, ISSN 1051-2004, (https://www.sciencedirect.com/science/article/pii/S1051200417302385)
>
> [2]A. Nguyen, J. Yosinski, J. Clune, Multifaceted feature visualization: uncovering the different types of features learned by each neuron in deep neural networks, CoRR, arXiv:1602.03616, 2016.
>
> [3] Mahendran, Aravindh, and Andrea Vedaldi. "Understanding deep image representations by inverting them." Proceedings of the IEEE conference on computer vision and pattern recognition. 2015.
>
> [4] A. Nguyen, J. Yosinski, Y. Bengio, A. Dosovitskiy, J. Clune, Plug & play generative networks: conditional iterative generation of images in latent space, CoRR, arXiv:1612.00005, 2016.
>
> [5]A. Nguyen, A. Dosovitskiy, J. Yosinski, T. Brox, J. Clune, Synthesizing the preferred inputs for neurons in neural networks via deep generator networks, in: Advances in Neural Information Processing Systems 29: Annual Conference on Neural Information Processing Systems 2016, Barcelona, Spain, 5–10 December, 2016, 2016, pp. 3387–3395.
>
> [6] M. Niepert, P. Minervini, and L. Franceschi. Implicit MLE: backpropagating through discrete exponential family distributions. Advances in Neural Information Processing Systems, 34:14567–
> 14579, 2021.
>
>
> ### Please kindly let us know if you have any concerns you find not fully addressed. We are more than happy to have a further discussion regarding it.

---

> > ### Comment · Reviewer_jhWR · 2022-11-23
> > **Answer to the authors**
> >
> > I thank the authors for their comprehensive answer. I think my comments were addressed satisfactorily and I have increased my evaluation to "Accept".

---

> ### Author Response · Authors · 2022-11-19
> **Response to Reviewer jhWR [2/3]**
>
> > In my experience, ... because it tends to "collapse" the answer on the reference point (the average of the embeddings). It would be interesting for the authors to evaluate whether this is also happening here (i.e., what happens if I simply take the term from (2) and visualize it after proper thresholding?).
>
> Thanks for your thoughtful comment. It might be true that collapsing to a reference point can lose some information, as mentioned in [1]. However, that is not the case in our work because the mean embedding computed over the training data encapsulates the representative features of this class perceived by the GNN model itself. Therefore, enforcing the explanation embedding to be closer to the mean embedding will not result in information loss for the purpose of interpreting the GNN model. Intuitively, we attempt to find an explanation graph that contains both the discriminative features ( enforced by the activation maximization term from equation 2) and the representative features (enforced by the cosine similarity term from equation 2) of this class perceived by the GNN model. Additionally, the cosine similarity term can also prevent the generated explanation from deviating from the true data distribution so that the explanation is more consistent with the true graphs. **Speaking of the experimental study, we presented an ablation study of the cosine similarity term in section 5.2 of the Manuscript.** This ablation study shows that the explanation without the cosine similarity term completely deviates from the true data distribution, even though its class score is much higher than the explanation with the cosine similarity term. This further demonstrates that the cosine similarity term can indeed help us to generate a more meaningful explanation that could potentially provide insights into how the model actually reasons over the true data distribution. All in all, we carefully design this novel model-level loss function to interpret the GNN for the sake of generating realistic explanation graphs with discriminative and representative features of that class perceived by the GNN model.
>
>
> > The core innovation of the work is using the GS trick to reparameterize the gradients. Also here, this is quite common today in the GNN literature (e.g., DGM for latent graph imputation, PGExplainer for explanation, adaptive graph sparsification, ...).
>
> Thanks for your constructive comment. We totally agree that adopting the Gumbel-Softmax trick (GS trick) in GNN literature is not a new idea. We are the first method that adopts the GS trick to provide model-level explanation for GNN, but we are not the first method adopting the GS trick in GNN literatures. We also acknowledge that there is a lack of discussion regarding the application of the GS trick in GNN, especially the PGExplainer, which is another method that adopts the GS trick to explain GNN. To be specific, PGExplainer trains an MLP to predict the explanation distribution for a graph instance and then reparameterize the explanation distribution with the GS trick to sample an explanation subgraph with respect to a given graph instance. The main difference between PGExplainer and our method is that we learn a model-level explanation distribution obtained by gradient ascent, while they learn an instance-level explanation distribution predicted by an MLP. Besides, our objective function is maximizing the activation and the cosine similarity between the explanation embedding and mean embedding, whereas their objective function is maximizing the mutual information between the GNN prediction of the explanation subgraph and the GNN prediction of the original graph.
>
> We would like to take this opportunity to emphasize the core innovation of this work. Our novelty is mainly about designing a new objective function for generating a meaningful model-level explanation for GNNs. Specifically, the activation maximization along with the cosine similarity term allows us to generate a more realistic explanation graph with the discriminative and representative features of that class perceived by the GNN model. The basic idea of our objective function design might seem to be straightforward, but it indeed has some clear advantages over the existing literatures in CV (i.e., AM+expert and AM+decoder) which also attempt to generate a more realistic-looking explanation image. These advantages include computational efficiency, the capability of interpreting GNNs, and the explainability of the GNNInterpreter itself. Compared with the state-of-the-art model-level explanation method for GNNs, we are more computationally efficient, more flexible in explaining GNNs in different application domains without requiring manually defined domain-specific rules, and more applicable to broader usage scenarios with different types of node features and edge features. We believe that this work has some unique value in terms of diagnosing and identifying the potential pitfalls of GNNs.

---

> ### Author Response · Authors · 2022-11-19
> **Response to Reviewer jhWR [1/3]**
>
> We sincerely appreciate your valuable time and efforts in reviewing this paper. We thank the thoughtful feedbacks you provided, which significantly improved the quality of this paper. For the potential concerns you bring up, we would like to answer/address them as the followings.
>
> > First, the method which is proposed is basically activation maximization applied to a graph, with a simple "localization" term (see, e.g., [1]) to force the embeddings to lie close to the embeddings of the real graphs. I would suggest to discuss explicitly the relation of the method with known methods from the CV literature.
>
> Thanks for your insightful comment. We strongly agree that the activation maximization term (the first term from equation 2) is the same as the model-level explanation methods of other deep learning models (e.g., CNN and DNN). However, the cosine similarity term (the second term from equation 2) we proposed for explaining GNN is new in this field. Next, we would like to discuss the relationship between our approach and three types of methods with activation maximization[1] in CV, respectively.
>
> In section 3.4 of [1], where the concept of “localization” is introduced,  [2] is mentioned because [2] tries to interpret a class using the local prototypes in the situation that the examples in the same class might be multimodal. For example, the classifier is trained to distinguish apples and bananas, but the apple class contains both green apples and red apples. In this scenario, [2] attempts to visualize different facets of the same class by mapping the embedding of all apples to 2-D space with t-SNE and running K-means clustering to manually identify different facets (green apple and red apple). Then, they initialize the activation maximization with the mean image of the corresponding facet instead of a random image. Compared to our method, [2] not only have a different motivation but also use a different approach. Their goal is to generate the local prototype for different facets in the same class, but our goal is to enforce the generated global prototype to be similar to the true graphs. In terms of the methodology, they first manually identify different facets of the same class with K-means clustering and then compute the mean image of different facets as initialization for activation maximization. In contrast, we minimize the cosine similarity between the explanation embedding and the mean embedding to generate graphs taking into account the domain-specific knowledge learned by the model.
>
>
> Undoubtedly, there are some existing works in CV that have similar goals as our method. Specifically, AM+expert (mentioned in section 3.2 of [1]) and AM+decoder (mentioned in section 3.3 of [1]) all attempt to generate a realistic-looking image to maximize the class activation. Their goal is similar to our goal because we also aim to generate the explanation graphs without deviating much from the true data distribution. However, our fundamental approach is different than AM+expert and AM+decoder. The existing works of AM+expert [3,4] train a model to learn the data density function. Then, they generate an image to maximize the class score and the log data likelihood predicted by the expert model simultaneously. Nonetheless, this expert model is often very hard to learn or can be a very complex model, which makes the maximization of log data likelihood become difficult. The existing works of AM+decoder train a decoder model (i.e., GAN[5]) to draw synthetic samples from the true data distribution such that the activation can be maximized.
>
> We take a completely different approach than the AM+expert and AM+decoder and also have several advantages over them. First, we are more computationally efficient because we do not require to train another deep learning model to inject the domain-specific knowledge. Secondly, our entire explanation pipeline does not have any mysterious parts left to be explained, whereas AM+expert and AM+decoder try to eliminate a black box by creating another one. Lastly, our approach is more appropriate than training another decoder model, considering that the ultimate goal is to interpret the GNN model. The reason is that we take advantage of the embedding function learned by the GNN itself to inject the domain-specific knowledge (extracted by the GNN itself) into the generated explanation graphs. In conclusion, compared with the existing localization methods in CV [2], our work is fundamentally different, in terms of both the methodology and the goal. Compared with the existing methods of generating realistic-looking explanation images in CV, we take a different approach with several advantages over them. **In Section 2 and Section 4.1 of the manuscript, we have clarified that the activation maximization term we have is similar to the model-level explanation methods for other deep-learning models.**

---

### Official Review · Reviewer_ErAv · 2022-10-25

**Confidence:** 4
**Correctness:** 3
**Technical Novelty And Significance:** 2
**Empirical Novelty And Significance:** 2
**Recommendation:** 8

**Clarity, Quality, Novelty And Reproducibility:**

**Clarity**

The paper is clearly written and provides a detailed motivation for the need for model-level explanations.

**Novelty**

The paper tackles generating model-level explanations for GNN predictor models which haven't been explored much.

**Reproducibility**

No code implementation details have been shared.

**Strength And Weaknesses:**

**Strengths**

1. The paper presents a new method for generating model-level explanations for graphs with different types of node and edge features.
2. The proposed algorithm is computationally efficient and takes 10x less time than currently used graph explainer algorithms.

**Weaknesses and Open Questions**
1. The paper argues that an important goal of a GNN model explainer is to confine the distribution of the explanation graphs to the domain-specific boundary and leverages the abstract knowledge learned by the GNN itself to prevent the explanation from deviating from the true data distribution. However, the distribution from this representation may not be essentially similar to the true data distribution.
2. GNNInterpreter assumes white-box access to the model and training dataset, where it approximates the domain knowledge using the average embedding of all graphs $\psi_{c}$ from the target class in the training set. This drastically limits the approach to smaller models and datasets.
3. The description of GNNInterpreter is incomplete in the main tex. For instance, Equation 8 only provides the generic loss for training $\omega_{ij}$, $\eta_{ij}$, and $\xi_{ij}$ parameters and does not talk about the critical $\ell_{p}$-loss and other constraints.
4. The paper does not talk about the *Redundant Evidence* pitfall mentioned in Faber et al. [1] Do they report the results by taking the maximum score across all possible explanations for a given class?
5. It would be great to comment on other baseline explainers for graph classifiers like [2-3] or a random baseline.

**References**
1. Lukas Faber, Amin K. Moghaddam, and Roger Wattenhofer. When comparing to ground truth is wrong: On evaluating gnn explanation methods. In KDD, 2021.
2. Veyrin-Forrer, L., Kamal, A., Duffner, S., Plantevit, M., & Robardet, C. On GNN explanability with activation patterns, 2021.
3. Duval, Alexandre, and Fragkiskos D. Malliaros. GraphSVX: Shapley value explanations for graph neural networks. In Joint European Conference on Machine Learning and Knowledge Discovery in Databases, Springee, 2021.


**Summary Of The Paper:**

With the increasing use of complex black-box Graph Neural Networks (GNNs) in high-stakes applications, it is critical to developing explanation algorithms to understand their decisions. To this end, several methods have been proposed to generate instance-level explanations for GNN predictors. However, most works focus on generating predictions for individual node- or graph-class predictions and fail to generate model-level explanations that aim to understand the global behavior of the model. In this work, the authors propose, GNNInterpreter, a probabilistic generative model-level explanation method for GNNs that learns an explanation graph distribution that identifies the discriminative features of a graph for a given class prediction. Experimental results show that GNNInterpreter generates faithful explanations for synthetic and real-world datasets.

**Summary Of The Review:**

Look at the strengths and weaknesses for complete details.

---

> ### Author Response · Authors · 2022-11-19
> **Response to Reviewer ErAv [4/4]**
>
> >It would be great to comment on other baseline explainers for graph classifiers like [2-3] or a random baseline.
>
> Thanks for your valuable suggestion. We totally agree that we have missed some important references in our paper, including the two explainers for graph classification you mentioned above.  **We have revised our related work section, which specifically mentions Inside-GNN[2] and GraphSVX[3].** Generally speaking, almost all the instance-level explanation methods for GNNs can be applied to explain graph classifiers. However, we have a completely different focus than the instance-level explanation methods because we are trying to open the black-box in a different way. Our method attempt to interpret the high-level decision-making process of GNNs, while the instance-level explanation method aims at explaining why a certain prediction is made for a specific graph instance. Unfortunately, due to the page limit, we are unable to discuss every instance-level explanation method in detail because most instance-level methods, in fact, are not very related to our method.
>
> GraphSVX [3] is also an instance-level explanation method, which proposes a surrogate model to learn the importance of nodes and the node features toward the GNN prediction of a specific graph instance. To learn this surrogate model, they first construct a perturbed dataset for that graph instance by jointly applying various combinations of the node masks and feature masks to the input graph. Then, they utilize the Weighted Linear Regression (WLR) as the surrogate model such that the coefficient of WLR can be interpreted as the Shapley Value measuring the inﬂuence of different node or node features on the GNN prediction of that graph instance. Compared with GraphSVX, we take a fundamentally different approach to accomplish an entirely different goal.
>
> Inside-GNN[2] is a very interesting work that focuses on studying the internal mechanism of GNNs. Specifically, they try to interpret the GNNs by analyzing the activation pattern in the hidden layers, unlike our method focusing on interpreting the relation between the input graph and the output layer. They mine the activation pattern in the hidden layers to accomplish two goals: (1) support instance-level explanation; (2) provide a set of activation patterns to summarize all graphs in the training data. Their first goal is very similar to the goal of other instance-level explanation methods including GraphSVX. Their second goal is accomplished by finding different sets of graphs that have similar activation patterns and generating a common subgraph to summarize the shared features for each set. Thus, all the common subgraphs jointly can be regarded as a summarization of different features in the training data. Inside-GNN has some unique value in interpreting the hidden layers of GNNs. However, if the ultimate goal is to examine the model reliability and to identify the potential model pitfall, our method might be a better choice. The reason is that we can synthetically generate the discriminative features of a given class perceived by the GNN model. If the discriminative features of the target class encapsulated in the explanation graph deviate from the expected discriminative feature, it indicates that the GNN model is cheating by using the incorrect discriminative feature to differentiate the target class and other classes. However, Inside-GNN just summarizes the different features of the graphs in the training data, which cannot provide insights about how GNN would behave on unseen graphs and thus cannot help to identify the potential model pitfall.
>
> Thanks to your suggestion, we also managed to compare our method with the random baseline. Specifically, for each classifier, we generate 1000 Gabriel random graphs with the same number of nodes as the average number of nodes in the training data. **Then, the average class probability with the standard deviation for 1000 random graphs is computed and reported in Table 5.** As you can see, for all the classifiers we explained, our method can obtain a significantly higher class probability than the random baseline.
>
> ## Reference
> [2] Veyrin-Forrer, L., Kamal, A., Duffner, S., Plantevit, M., & Robardet, C. On GNN explanability with activation patterns, 2021.
>
> [3] Duval, Alexandre, and Fragkiskos D. Malliaros. GraphSVX: Shapley value explanations for graph neural networks. In Joint European Conference on Machine Learning and Knowledge Discovery in Databases, Springee, 2021.
>
> ### Please kindly let us know if you have any concerns you find not fully addressed. We are more than happy to have a further discussion regarding it.

---

> ### Author Response · Authors · 2022-11-19
> **Response to Reviewer ErAv [3/4]**
>
> >The description of GNNInterpreter is incomplete in the main text. For instance, Equation 8 only provides the generic loss for training ωij, ηij, and ξij parameters and does not talk about the critical ℓp-loss and other constraints.
>
> Thanks for your invaluable suggestions. **We have revised Section 4.3 of the manuscript to describe our regularization constraint.**
>
> >The paper does not talk about the Redundant Evidence pitfall mentioned in Faber et al. [1] Do they report the results by taking the maximum score across all possible explanations for a given class?
>
> Thanks for raising this interesting point. The redundant evidence pitfall mentioned in [1] is a potential problem in evaluating the generated explanation when the ground truth explanations are not unique.  Specifically, when there are multiple ground truth explanations, comparing the generated explanation against only one ground truth explanation might yield wrong weak results. However, for a model-level explanation method, the ground truth explanation graph does not exist because we never know what the true discriminative feature the GNNs learned for each class is. Therefore, we don’t rely on the ground truth explanation to evaluate our method. Instead, we can directly measure how discriminative the generated explanation is for a certain class by the corresponding class score predicted by the GNN. If the class score after softmax is close to 1, it indicates that the generated explanation indeed contains the discriminative features GNN tries to detect for a certain class. In a word, the redundant evidence pitfall in evaluating our method does not exist because we don’t rely on the ground truth explanation to evaluate our method.
>
> It is worth mentioning that there might exist multiple discriminative features the GNN tries to detect for a specific class. In this case, different explanation graphs could be generated by applying our method for multiple times with different random initialization. Given that generating an explanation per class only takes less than a minute (as shown in Table 5), we believe generating multiple explanations per class is feasible in terms of the time cost. **We reported multiple explanations per class for all datasets in Figure 8 from Appendix F for your reference. We also briefly discussed the redundant evidence pitfall in Appendix F.**
>
> ## Reference
> [1] Lukas Faber, Amin K. Moghaddam, and Roger Wattenhofer. When comparing to ground truth is wrong: On evaluating gnn explanation methods. In KDD, 2021.

---

> ### Author Response · Authors · 2022-11-19
> **Response to Reviewer ErAv [2/4]**
>
> >GNNInterpreter assumes white-box access to the model and training dataset, where it approximates the domain knowledge using the average embedding of all graphs ψc from the target class in the training set. This drastically limits the approach to smaller models and datasets.
>
> Thanks for your thoughtful comment. The main goal of this work is to examine the model reliability (e.g., identifying bias attribution issues, potential risk of classifying unseen graphs with misleading graph patterns, etc.) instead of explaining why certain prediction is being made for a specific input instance. The people who use this method will mainly be the model developer instead of the end-user because it is the developer’s job to ensure the model is reliable. Therefore, it is safe to assume we have white-box access to the model parameter and the training data.
>
> Regarding the concern about the model size and data size, we fully understand where this concern comes from because we need to compute a mean embedding for optimization purposes. However, we don’t think this could be a potential weakness of this method. For the large dataset, we just need to compute the mean embedding once over the entire training data (assuming we want to explain all classes). The computational cost of computing the mean embedding for all the classes is even less than the cost of training GNN for one epoch because we will only perform one forward pass over the entire training data. Besides, the computational cost of computing the mean embedding can be further reduced by bootstrapping a random subset of training data. Speaking of the limitation on model size, since we perform gradient descent to optimize the loss function, our method is capable of interpreting the GNN model in different sizes as long as you can perform the back-propagation over the GNN model.
>
> **Lastly, we also include the time complexity analysis of our method and XGNN in the Appendix B.** Our time complexity is $O(TKN^2)$, where $T$ is the number of iterations, $K$ is the number of Monte-Carlo samples, $N$ is the number of vertices for the Gilbert random graph distribution (the max number of nodes in the explanation graph). The time complexity of XGNN is $O(TRM^3)$, where $T$ is the number of episodes, $R$ is the number of Rollouts, and $M$ is the maximum number of edges in the explanation graph. For a connected explanation graph with $N$ nodes and $M$ edges, $M > N-1$ always holds. Therefore, $O(TRM^3)$ is much higher than $O(TKN^2)$, since $R$ and $K$ are constant hyper-parameters. Our method outperforms XGNN in terms of time complexity.  Additionally, according to the quantitative evaluation result in Table 5, it only takes less than a minute for our method to explain one class in a GNN model, which is as many as 10 times faster than XGNN. In conclusion, our approach is capable of explaining large models with large training data, at a low computational cost.

---

> ### Author Response · Authors · 2022-11-19
> **Response to Reviewer ErAv [1/4]**
>
> We sincerely appreciate your valuable time and efforts in reviewing this paper. We thank the thoughtful feedback you provided, which significantly improved the quality of this paper. For the potential concerns you bring up, we would like to answer/address them here.
>
> >The paper argues that an important goal of a GNN model explainer is to “confine the distribution of the explanation graphs to the domain-specific boundary and leverages the abstract knowledge learned by the GNN itself to prevent the explanation from deviating from the true data distribution”. However, the distribution from this representation may not be essentially similar to the true data distribution.
>
> Thanks for your insightful comment. Statistically speaking, the graphs of class c in the training data can be regarded as samples of the entire population of graphs in this class. We can never know the true data distribution because the population might be infinite. We can only approximate the true data distribution using a sample of graphs (i.e., the training data). According to the Law of Large Numbers, the larger the sample size, the closer the sample mean is to the population mean. Therefore, maximizing the cosine similarity between the explanation embedding and the mean embedding of training data (the sample mean) can be an effective way to prevent the explanation from deviating from the true data distribution.
>
> The question is how close the explanation graph distribution will be to the true data distribution. The size of the training data will be one of the important factors. The larger the training data, the closer the explanation distribution is to the true data distribution (Law of Lage Numbers). Another important factor is whether the training data is an unbiased sample of the population of all the true graphs in this class.  If the samples we collect as the training data are unbiased, the sample mean should be very close to the population mean. Suppose the training data is biased samples. In this case, the explanation graph distribution may not be close to the true data distribution because the training data distribution will deviate from the true data distribution to some extent. Then, the GNN model may not generalize well for the true data distribution. However, the biased samples will not compromise the faithfulness of our explanation method because we can still generate faithful explanations in the domain where this model is clearly defined during training. In terms of explaining the model behavior in a domain far away from the training data distribution, it is not the main focus of this work but still might be meaningful in some real-world scenarios. You can always substitute the mean embedding of training data with the mean embedding of another set of graphs. For example, if you build a classifier to distinguish apples and bananas, but your training data only contains red apples. Your goal is to interpret the model behavior on green apples. To obtain that explanation graph, you can just compute the mean embedding over a set of green apples and feed that into our method. Then, our method will attempt to generate an explanation that contains the discriminative features of apples (enforced by the first term in equation 2) and resembles the representative features of green apples (enforced by the second term in equation 2). We note that both the discriminative and the representative features are captured and encapsulated by the GNN itself, which allows us to better understand the high-level working mechanism inside the GNN.

---

### Comment · Area_Chair_yKNV · 2022-11-23
**Rebuttal, please become active**

Dear reviewers,

The authors have made an effort to write rebuttals to your reviews. Please read, think, and respond to the rebuttals. As we all know, most authors put a lot of effort into their work and the rebuttals, so the least we should do is acknowledge the responses.

Thank you.

AC

---

### Author Response · Authors · 2022-11-30
**Thanks to Reviewers and AC**

Dear Reviewers and AC,

We would like to sincerely thank you for your invaluable time and effort in reviewing this paper. Your constructive suggestions and insightful comments significantly improved the quality of this paper. We fully enjoyed the thorough discussion with you in the past few weeks, and some comments even pointed us to an interesting direction for our future research. We are more than happy to answer any questions you may have later.

Paper 6066 Authors

---

### Decision · Program_Chairs · 2023-01-20

**Decision:**

Accept: poster

**Justification For Why Not Higher Score:**

There were still some weak points identified by the reviewers. Especially one reviewer only gave a score marginally above the acceptance threshold. Hence, a poster presentation seems appropriate.

However, this paper could also be a candidate for a spotlight presentation.

**Justification For Why Not Lower Score:**

All reviewers gave scores above the acceptance threshold.

**Metareview: Summary, Strengths And Weaknesses:**

The authors propose a model-based explanation method for the class of GNNs following a message-passing scheme. There was a thorough and lively discussion phase during which the authors provided enough improvements and clarifications to satisfy all reviewers. This is a clear accept.

**Note From Pc:**

if the above contains the word "oral" or "spotlight" please see: "oral" presentation means -> notable-top-5% and "spotlight" means -> notable-top-25%. As stated in our emails, we are disassociating presentation type from AC recommendations